# PROACTIVE AGENTS FOR MULTI-TURN TEXT-TO-IMAGE GENERATION UNDER UNCERTAINTY

## ABSTRACT

User prompts for generative AI models are often underspecified or open-ended, which may lead to sub-optimal responses. This prompt underspecification problem is particularly evident in text-to-image (T2I) generation, where users commonly struggle to articulate their precise intent. This disconnect between the user's vision and the model's interpretation often forces users to painstakingly and repeatedly refine their prompts. To address this, we propose a design for proactive T2I agents equipped with an interface to actively ask clarification questions when uncertain, and present their understanding of user intent as an interpretable *belief graph* that a user can edit. We build simple prototypes for such agents and verify their effectiveness through both human studies and automated evaluation. We observed that at least 90% of human subjects found these agents and their belief graphs helpful for their T2I workflow. Moreover, we use a scalable automated evaluation approach using two agents, one with a ground truth image and the other tries to ask as few questions as possible to align with the ground truth. On DesignBench, a benchmark we created for artists and designers, the COCO dataset (Lin et al., 2014) and ImageInWords (Garg et al., 2024), we observed that these T2I agents were able to ask informative questions and elicit crucial information to achieve successful alignment with at least 2 times higher VQAScore (Lin et al., 2024) than the standard single-turn T2I generation. Demo: `https://youtu.be/HPgJ4xPRnto`

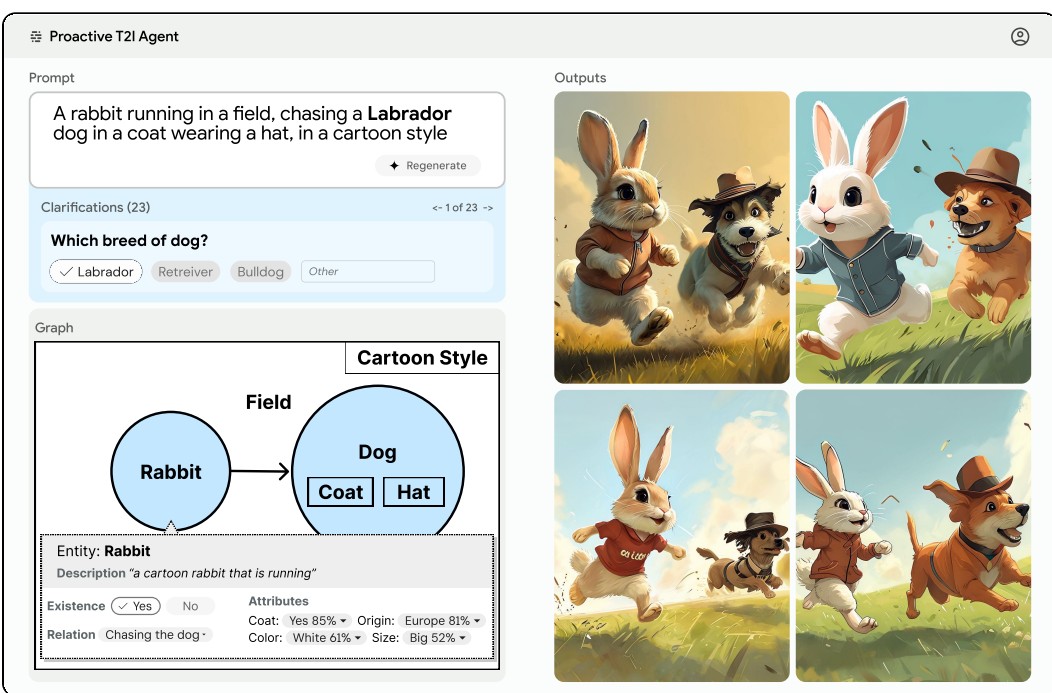

Figure 1: A proactive text-to-image agent interface that clarifies prompts, updates them based on user responses, and expresses its uncertainty and understanding as an editable belief graph.

# 1 INTRODUCTION

A fundamental challenge in the development of AI agents is how to foster effective and efficient multi-turn communication and collaboration with human users to achieve user-defined goals, especially when faced with the common issue of vague or incomplete instructions from humans. We focus specifically on text-to-image (T2I) generation, where recent advancements (Baldridge et al., 2024; Betker et al., 2023; Podell et al., 2023; Yu et al., 2023) have enabled the creation of stunning images from complex text descriptions. However, users often struggle to describe the image they would like to generate in a way that T2I systems can fully understand. This leads to unsatisfactory results and repeated iterations of prompts.

The prompt underspecification problem arises from the inherent ambiguity of natural language, the different assumptions that humans make and the vast space of potential images that can be generated from a single prompt (Hutchinson et al., 2022). Imagine a prompt *generate an image of a rabbit next to a cat*. This seemingly simple prompt leaves many important aspects underspecified: *What kind of rabbit? What color is the cat? What is their relative positions? What is the background?* While a T2I model can generate an image with a rabbit and a cat in it, it is unlikely that the image captures the specific details a specific user has in mind. For example, people in Holland might assume it is common for rabbits to have lop ears, but people in New England might expect to see cottontail rabbit with straight ears. The combination of all these factors can lead to a frustrating cycle of trial-and-error, with the user repeatedly refining their prompt in an attempt to steer the model towards the desired output (Vodrahalli & Zou, 2024; Huang et al., 2024; Sun & Guo, 2023).

Instead of relying on passive T2I models that simply generate images based on potentially vague user instructions, we pursue a quest for agency in T2I generation. The T2I agents should actively engage with human users to provide a collaborative and interactive experience for image creation. We envision that these T2I agents will be able to (1) express and visualize their beliefs and uncertainty about user intents, (2) allow human users to directly control their beliefs beyond just text descriptions, and (3) proactively seek clarification from the human user to iteratively align their understanding with what the human user intends to generate.

In this work, we develop simple prototypes of such agents. At the core of those agent prototypes, we build in a graph-based symbolic belief state, named *belief graph*, for agents to understand its own uncertainty about possible entities (e.g., rabbit) that might appear in the image, attributes of entities (e.g., rabbit's color), relations between entities and so on. Given a user prompt, we use an LLM and constrain its generation to the graph structure of beliefs, which include probability estimates on the appearance of entities and the possible values for attributes and relations. Figure 1 illustrates the interface and features of the prototypes. In particular, the agent can ask questions based on its uncertainty. For example, a very simple strategy is to find the most uncertain attribute of an entity (e.g., rabbit's color) and use an LLM to phrase a question about the attribute (e.g., What is the color of the rabbit?). The agent can also guide users to directly edit items in the graph.

To evaluate the utility of our agent prototypes, we conduct both human studies and automatic evaluations. The human studies aim to understand how helpful simple T2I agents can be, and evaluate how good the agents' questions are. We develop automatic evaluation pipelines to assess the effectiveness and efficiency of the T2I agents when interacting with simulated users with underspecified prompts answering questions based on their pre-fixed intents.

We found that over 90% human subjects expect proactive clarifications to be helpful, and 58% think this question asking feature of agents could deliver value to their work very soon, or immediately. We create a hand-curated benchmark called DesignBench which contains aesthetic scenes with multiple entities and interactions between entities; it also contains both a short and long caption. DesignBench also features diversity between photo-realism, animation and multiple styles allowing a robust testing with the use case of artists and designers in mind. This benchmark will be released with this paper. We run automatic evaluations on both the COCO dataset (Lin et al., 2014) and DesignBench. We found that our agents can achieve at least 2 times higher VQAScore (Lin et al., 2024) than the traditional single-turn T2I generation within 5 turns of interaction.

Our contributions: (1) the first interpretable and controllable belief graph used for T2I, (2) novel design and prototypes for T2I agents that adaptively ask clarification questions and present belief graphs; (3) a new automatic evaluation pipeline with simulated users to assess question-asking skills of T2I agents; and (4) DesignBench: a new T2I agent benchmark. Appendix A details the novelty.

## 2 RELATED WORK

From the very outset of **artificial intelligence**, a core challenge has been to develop intelligent agents capable of representing knowledge and taking actions to acquire knowledge necessary for achieving their goals (McCarthy & Hayes, 1969; Minsky, 1974; Moore, 1985; Nilsson, 2009; Russell & Norvig, 2016). Our work is an attempt to address this challenge for intelligent T2I agents.

In **machine learning and statistics**, efficient data acquisition has been extensively studied for many problems, including active learning (Settles, 2009), Bayesian optimization (Garnett, 2023), reinforcement learning (Kaelbling et al., 1996; Sutton, 2018) and experimental design (Chaloner & Verdinelli, 1995; Kirk, 2009). We reckon that T2I agents should also be capable of actively seeking important information from human users to quickly reduce uncertainty (Wang et al., 2024b) and generate satisfying images. In §D, we detail the implementation of action selection strategies for our T2I agents.

In **human-computer interaction**, researchers have been extensively studying how to best enable Human-AI interaction especially from user experience perspectives (Norman, 1994; Höök, 2000; Amershi et al., 2019; Cai et al., 2019; Viégas & Wattenberg, 2023; Chen et al., 2024; Yang et al., 2020; Kim et al., 2023). Interface design for AI is becoming increasingly challenging due to the lack of transparency (Viégas & Wattenberg, 2023; Chen et al., 2024), uncertainty about AI capability and complex outputs (Yang et al., 2020). We aim to build user-friendly agents, and an indispensable component is their interface to enable them to effectively act and observe, as detailed in §G.

**Interpretebaility.** Surfacing an agent's belief overlaps with interpretability as both aim to understand model or agent's internal. Some methods leverage LLM's natural language interface to surface their reasoning (e.g., chain of thought (Wei et al., 2023a)), sometime interactively (Wang et al., 2024a). While these approaches makes accessible explanations, whether the explanations represents truth has been questioned (Lanham et al., 2023; Wei et al., 2023b; Chen et al., 2023). Some studies indicate explanations generated by the LLMs may not entail the models' predictions nor be factually grounded in the input, even on simple tasks with extractive explanations (Ye & Durrett, 2022).

**Text-to-Image (T2I) generation.** Text-to-image prompts can be ambiguous, subjective (Hutchinson et al., 2022), or challenging to represent visually (Wiles et al., 2024). Different users often have distinct requirements for image generation, including personalization (Wei et al., 2024), style constraints (Wang et al., 2023), and individual interpretations (Yin et al., 2019). To create images that better align with users' specific needs and interpretations, it is essential to actively communicate and interact with the user to understand the user's intent.

**Multi-turn T2I.** Current multi-turn T2I systems typically focus on multi-turn user instructions. Huang et al. (2024); Sun & Guo (2023) propose multi-modal interactive dialogue systems which passively respond to user's natural language instructions. Mini DALL·E 3 (Lai et al., 2023) builds an interactive T2I framework with an LLM in the loop to have a dialogue with the user via text chat and improve image generation and editing based on the entire conversation. Vodrahalli & Zou (2024) collected and analyzed a dataset of human-AI interactions where users iteratively refine prompts for T2I models to generate images similar to goal images (goal images are only visible to users). This may require users to actively try prompts to understand model behaviors. On the contrary, our work aims to reduce the burden on the user by actively asking questions to understand user intents.

A core challenge in multi-turn T2I is consistency (Cheng et al., 2024a;b; Zeqiang et al., 2023). Hu et al. (2024) introduce Instruct-Imagen, which is a model that follows complex multi-modal instructions. AudioStudio (Cheng et al., 2024a) is a multi-turn T2I framework aimed at subject consistencies while generating diverse and coherent images. These consistency improvement methods can be integrated into T2I agents but it is beyond the scope of this work. Our key focus is on the sequential decision making capability of agents to elicit user intents.

## 3 BACKGROUND

The belief graph in our work is closely related to symbolic world representations.

**World states.** In classical AI, researchers use symbolic representations to describe the world state (McCarthy & Hayes, 1969; Minsky, 1974; 1988; Pasula et al., 2007; Kaelbling & Lozano-Pérez, 2011). For example, in the blocks world (Ginsberg & Smith, 1988; Gupta & Nau, 1992; Alkhazraji

et al., 2020), a state can be

$$is\_block(a) \land is\_red(a) \land on\_table(a) \land is\_block(b) \land is\_blue(b) \land on(b, a),$$

describing that there are a red block and a blue block, referred to as $a$ and $b$, block $a$ is on a table, and block $b$ is on $a$. Such world states must include **entities** (e.g., $a$ and $b$), their **attributes** (e.g., position $on\_table$, characteristics $is\_block$) and **relations** (e.g., $on(b, a)$) which are critical for enabling a robot to know and act in the world.

In linguistics, Davidson (1965; 1967b;a) introduce logic-based formalisms of meanings of sentences. The semantics of a sentence is decomposed to a set of atomic propositions, such that no propositions can be added or removed from the set to represent the meaning of the sentence. (Cho et al., 2023) propose Davidsonian Scene Graph (DSG) which represent an image description as a set of atomic propositions (and corresponding questions about each proposition) to evaluate T2I alignment.

We borrow the same concept as symbolic world representations and scene graphs, except that the agent needs to represent an imaginary world. The image generation problem can be viewed as taking a picture of the imaginary world. The world state should include all entities that are in the picture, together with their attributes and relations.

**Belief states.** Term "belief state" (Nilsson, 1986; Kaelbling et al., 1998) has been used to describe a distribution over states. E.g., for block $a$, we might have $p(on\_table(a)) = 0.5$ and $p(\neg on\_table(a)) = 0.5$, which means the agent is unsure whether the block is on a table. To represent the T2I agent's belief on which image to generate, we need to consider the distribution over all possible "worlds" in which the picture can be taken. This distribution can be described by the probabilities that an entity appears in the picture, an attribute gets assigned a certain value, etc.

## 4 PROACTIVE T2I AGENT DESIGN

We provide high-level principles and design that guide our agent how to behave and interact with users to generate desired images from text through multi-turn interactions. The goal of the agent is to generate images that match the user's intended image as closely as possible with minimal back-and-forth, particularly in cases with underspecified prompts and the agent needs to gather information proactively. This requires a decision strategy on information gathering to trade off between the cost of interactions and the quality of generated images. The formal problem definition can be found in §B.

We equip the agent with the ability to gather information in two ways: ask clarification questions (§4.1) and express its uncertainty and understanding in a way that users can edit (§4.2). Once a piece of information is collected from a user, the agent also need to update its questions and uncertainty (§4.3). To enable all these agent behaviors, we need to situate the agent in an interface to effectively communicate with users (§G). In the following, we introduce the design of the above components under the interface, to ensure information efficiency for T2I generation.

### 4.1 WHAT KIND OF QUESTIONS SHOULD BE ASKED?

We explain considerations in question asking and examples of strategies in this section.

#### 4.1.1 PRINCIPLES

We identify the following principles for an agent to ask the user questions about the underspecified prompt and their intended image: (i) **Relevance**: The question should be based on the user prompt. (ii) **Uncertainty Reduction**: The question should aim to reduce the agent's uncertainty about the attributes and contents of the image, the objects, the spatial layout, and the style. (iii) **Easy-to-Answer**: The question should be as concise and direct as possible to ensure it is not too difficult for the user to answer. (iv) **No Redundancy**: The question should not collect information present in the history of interactions with the user. The Relevance and No Redundancy principles are self-explanatory, we detail the other two principles below.

**The Uncertainty Reduction principle** aims to let agent elicit information about various characteristics of the desired image, which the agent is unsure of.

First, the agent needs to know what characteristics of images are important. Some examples include: (i) Attributes of the subjects, such as breed, size, or color, with questions like *What kind of rabbit?*

*What color is the cat?*; (ii) Spatial relationships between the subjects, such as proximity and relative position (*Are the rabbit and cat close to each other? Are they facing each other?*); (iii) Background information, such as location, style and time of day (*Are they in a park or at home?*); and (iv) Implicit entities that might not be explicitly mentioned in the initial prompt but are relevant to the user's vision (*Are there any other animals or people present?*).

Second, the agent needs to know its own uncertainty about those characteristics. In the agent's belief, the uncertainty is explicit. One strategy is to form questions about the image characteristics that the agent is most uncertain about. We discuss more in §D.2.

Third, the agent needs to update its own uncertainty once the user gives a response to its question (a.k.a. transition in §4.3). Then, it can construct questions again based on its updated uncertainty estimates. This iterative clarification process allows the agent to progressively refine its understanding of the user's intent and generate an image that more accurately reflects their desired output.

**The Easy-to-Answer principle** aims to reduce users' effort to respond to questions. One way is to have the agent provide some answer options, where options are what the agent believes likely to appear. E.g., *What color is the cat? (a) Black (b) Brown (c) Orange (d) Other (please specify)*.

### 4.1.2 EXAMPLES OF QUESTION-ASKING STRATEGIES

Given the agent belief constructed from the user prompt (more details in §4.2), several basic approaches can be employed following the above principles. We construct simple agents with the following strategies, which are implemented and used in our experiments.

• Ag1 (§D.5): Rule-based question generation, which leverages predefined rules or heuristics to identify salient attributes, entities, or relationships that require clarification. For example, an LLM could be used to estimate the importance and likelihood of different components within the belief, and a heuristic could be applied to prioritize the most crucial elements for questioning.

• Ag2 (§D.6): Belief-guided question generation, which involves using natural language to represent the current understanding encapsulated in the belief. This representation, along with the conversation history, is provided as input to an LLM, guiding it to generate clarification questions.

• Ag3 (§D.7): Direct question generation, which write the above question-asking principles in a prompt for an LLM to generate a question.

### 4.2 INTERACTING WITH THE USER BASED ON AGENT BELIEFS

The Uncertainty Reduction principle inspires the usage of belief graphs for the agent to directly express uncertainty, in addition to reflecting uncertainty through questions. Instead of using hardcoded symbols in classic belief representations (Fikes & Nilsson, 1971) described in §3, we employ LLMs to generate names and values for entities, attributes and relations. As a result, this belief construction method can generalize across any prompts. Algorithm 1 summarizes how we parse from a prompt to a belief graph and allow user interaction[1]. All agents in §4.1.2 use the same kind of belief graphs.

**Entities.** In addition to (a) entities mentioned in the user prompt, a belief graph also includes (b) implicit entities not mentioned in the prompt but likely to appear, e.g., *pet owner* in the context of a pet-related scene; and (c) background entities, such as *image style, time of day, location*, which play important roles in constructing the image.

---

**Algorithm 1** Belief Parsing and interaction

1: **Input:** Initial Prompt (IP)
2: **Initialization:** Merged Prompt (MP) ← IP
3: **for** $turn \leftarrow 1$ **to** $max\_turn$ **do**
4:     Parse entities from MP (D.8)
5:     Parse entity attributes and relations from entities and MP (D.9, D.10)
6:     Display belief graph, and collect interaction feedback (F)
7:     Update MP: MP ← MP + F (D.12)
8: **end for**

---

**Attributes and relations.** While the prompt might mention some attributes of a certain entity, they are not enough to describe the exact details of that entity. Hence the agent have to imagine the relevant attributes for each entity, and construct a list of possible values along with their associated probabilities (e.g., the *color* attribute for the *cat* entity might have values like *black, white, gray* with corresponding probabilities). Similarly the agent may have to imagine the possible relations between entities, e.g., *spatial relation* between *rabbit* and *cat* might include values like *close, far, touching*.

---
[1]The clarification question part of the interaction is omitted for simplicity

**Importance scores.** While the agent can be uncertain about many aspects of the user's intended image, some are more important than others. E.g., for prompt "a rabbit and a cat", the agent might be very uncertain about the exact color of a carpet that might appear in the image, but *rabbit* and *cat* are more important than the carpet. We enable agents to estimate an importance score for each entity, attribute and relation.

**Extracting beliefs and enabling interactions.** A simple idea is to use a large language model (LLM) via in-context learning. §D.1 details how an LLM may analyze the user prompt to identify entities, their attributes, and the relations between them, effectively translating the natural language input into a structured representation within the belief. Once the belief is extracted, a user can edit the belief to adjust uncertainty levels, confirm existence of entities etc, as shown in Figure 1.

### 4.3 TRANSITION

The agent belief undergoes a transition whenever the agent receives new information through user feedback, either user answers from the agent question or user interactions with the graph-based belief interface (Figure 1). This transition process integrates information from the initial user prompt, the conversation history, interaction and the previous belief to generate an updated belief of the user's desired image. Two possible approaches include: (i) Generate a comprehensive prompt that summarizes all interactions and information gathered thus far. This merged prompt is then used to re-generate the belief, effectively incorporating the new information into a refreshed representation. (ii) Leverage natural language to describe the accumulated information, including the initial prompt, conversation history, and user interactions. This descriptive summary is then provided as input to an LLM, instructing it to generate an updated belief based on the provided context. We use (i) for all agents in §4.1.2 and the implementation details can be found in §D.3.

## 5 EXPERIMENTS

We conduct 2 types of experiments to study the effectiveness of the proposed agent design: **automatic evaluation** which uses a simulated user to converse with a T2I agent and **human study** which studies the efficacy of our framework with human subjects.

### 5.1 AUTOMATIC EVALUATION

We simulate the user-agent conversation using self-play (Shah et al., 2018) between two LLMs. The conversation starts with an arbitrarily chosen image to represent the goal image from a T2I model that the user has in mind[2]. Along with this ground truth image, a user has a *detailed* prompt in mind that describes the image in high-detail. We use the algorithm similar to *Ag2* (detailed in §D.4) to simulate the user, where the questions are answered based on the ground truth prompt and the belief graph generated from the ground truth prompt. We run the agent-user conversation for a total of 15 turns[3] and compute different metrics at the end of each turn. More details of the simulated user can be found in the appendix, including the prompts provided to the LLM when simulating the user are provided. Figure 2 part b shows the multi-turn set up that we use in our results.

#### 5.1.1 SETUPS FOR AGENTS AND BASELINE

**Baselines.** We use a standard T2I model as a baseline, which directly generates an image based on a prompt without asking any questions. We refer to this baseline as 'T2I'.

**Agents.** We use Ag1, Ag2 and Ag3 with question-asking strategies introduced in §4.1.2. The creation and updates to the belief graph (§4.2), as well as transitions to prompt (§4.3) are consistent among all multi-turn agents. Further implementation details of each agent can be found in §D.

**Model Selection.** In this work we use an off-the shelve Text-to-Image (T2I) model and a Multi-Modal Large Language (MLLM) model and build the different components of our agent on top of these models. We keep these models consistent across all agents for fair comparison. We implement the agent on top of the Gemini 1.5 (Gemini Team Google, 2024) using the default temperature and a 32K context length. The in-context examples and the exact prompt used at each step of the agent

---

[2]This assumption only applies to the experiments. In practice, users don't necessarily have an image in mind, but they can get inspirations from the belief graphs and questions.

[3]While 15 turns is a suggested approximation of interaction time, accounting for varying difficulty between images, any number of turns can be used with this evaluation approach.

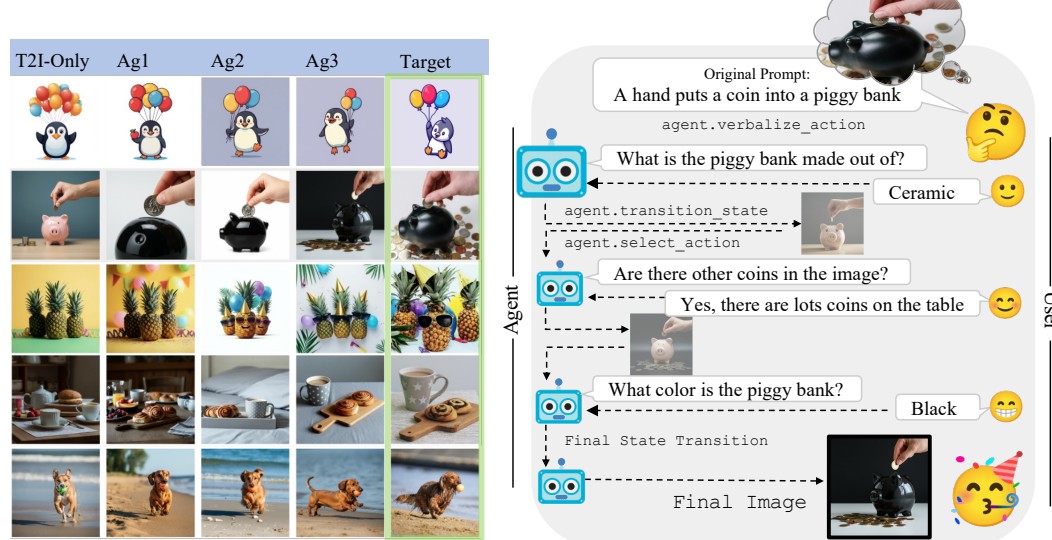

a) generated outputs and target image  b) multi-turn Ag3 example – real generated outputs

Figure 2: **a)** Each column displays the output of an agent after 15 turns - the right most column shows target image. Target images are part of DesignBench. **b)** A visualization of the multi-turn set up in the experiments. These are real generated outputs and simulated user outputs at turns 3, 10 and 15.

pipeline is detailed in §D.8 - §D.15. More agent implementation details are provided in §D. For T2I generation, we use Imagen 3 (Baldridge et al., 2024) across all baselines given it's recency and prompt-following capabilities. We used both the models served publically using the Vertex API[4].

### 5.1.2 DATASETS.

Our multi-turn agents aim to facilitate the generation of complex images, a process that often requires users to iteratively refine text-to-image (T2I) prompts until the generated image aligns with their mental picture. To evaluate these agents, we curate datasets comprising complex scenes involving multiple subjects, interactions, backgrounds, and styles. Each dataset consists of tuples: $(\mathbf{I}, p_0, c, b_{gt})$, where $\mathbf{I}$ represents the target image, $p_0$ is an initial (basic) prompt describing only the primary elements of the scene, $c$ is a ground truth caption providing a detailed description of $\mathbf{I}$, including spatial layout, background elements, and style, and $b_{gt}$ is the ground truth belief graph constructed via parsing $c$. The initial prompt $p_0$ is intentionally less detailed than $c$ to necessitate multi-turn refinement. This framework allows us to assess the agent's ability to guide the user towards the target image $\mathbf{I}$ starting from a simplified prompt.

Existing image-caption datasets primarily focus on simple scenes (Deng et al., 2009; Krizhevsky et al., 2009; Deng, 2012) or focus on very specific categories (Liu et al., 2016; Liao et al., 2022). With the aim for complex realistic images for testing the robustness of the Agents, we evaluate over the validation split of the Coco-Captions dataset (Chen et al., 2015). Five independent human generated captions are provided for each image in the dataset. These captions are often short and describe the basic elements contained in the image and the interactions between objects or persons in the image. We therefore select the shortest of the five human-generated captions and use this as a *starting prompt* $p_0$. We then use Gemini 1.5 Pro to expand the starting prompt by adding more details of the attributes of the entities in the image as well as the style and image composition which results in the *ground truth caption*. We also use the ImageInWords (Garg et al., 2024) dataset which takes a diverse set of realstic and cartoon images and has human annotators create dense detailed captions that describe attribute and relationships between objects in the image. In ImageInWords evaluations we use the long human annotation as the ground truth caption.

---

[4]https://cloud.google.com/vertex-ai

While COCO-Captions and ImageInWords provide complex, real-world images across diverse backgrounds, it lacks the artistic and non-photorealistic imagery often desired by designers and artists seeking to generate content outside the distribution of typical training data. To better evaluate our target for flexible use cases such as by artists, we introduce **DesignBench**, a novel dataset comprising 30 scenes specifically designed for this purpose. Each scene follows the $(\mathbf{I}, p_0, c)$ format described earlier. DesignBench includes a mix of cartoon graphics, photorealistic yet improbable scenes, and artistic photographic images. Examples from DesignBench and a comparison with COCO-Captions are provided in the Appendix.

### 5.1.3 METRICS

The outputs produced by the agent include a final generated image, a final caption and a final belief graph. We evaluate the agents across these modalities and evaluate their alignment to the ground truth image $\mathbf{I}$, $c$ and $b_{gt}$, using the following metrics.

**Text-Text Similarity**: We use 2 metrics for comparing the ground truth caption and the generated caption: 1) **T2T** – embedding-similarity computed using Gemini 1.5 Pro[5] and 2) **DSG** (Cho et al., 2024) adapted to parse text prompts into Davidsonian scene graph using the released code.

**Image-Image Similarity (I2I)**: We compute cosine similarity between the groundtruth image and the generated image from the agent prompt. We use image features from DINOv2 (Oquab et al., 2024) model following prior works.

**Text-Image Similarity**: We compare the ground truth prompt with the generated image (**T2I**) using the VQAScore (Lin et al., 2024) metric. We use the author released implementation of the metric and use Gemini 1.5 Pro as the underlying MLLM. More details about the T2I metrics can be found in §E.

**Negative log likelihood (NLL)**: We construct the ground truth state of the image in the form of a belief graph but with no uncertainty. We then approximately compute the NLL of the ground truth state given the belief of the agent at each turn, by assuming the independence of all entities, attributes and relations, and summing their log probabilities[6].

## 5.2 RESULTS FROM AUTOMATED EVALUATION

The results from the automatic evaluations in Table 1 show the $\mathbf{I}$, $c$ and $b_{gt}$ against each agents final generated image, text and state. All show the mean and standard deviation of the similarity metric at the final agent state. The blue row shows the baseline method which performs no updates to the prompt and instead applies the T2I model to the first prompt. Therefore this baseline represents the lower bound performance.

To add quantitative validity to the ground truth caption generation we perform Text to Image (VQA) Similarity between the ground truth caption and the ground truth over all images in the DesignBench dataset. The mean T2I VQA similarity between the ground truth caption and ground truth image is 0.99999985 with a median 1.0, and standard deviation of 4.5e-07. The mean is extremely close to 1 as expected of an accurate and well formed caption. These numbers can be compared to the T2I column of Table 1 to observe the delta between the ground truth caption and generated captions.

The results in Table 1 show that significant gains in performance come from using proactive multi-turn agents. The blue row shows the simplest baseline which directly uses a T2I model and performs no updates to the initial prompt $p_0$. We see that all of the multi-turn agents far exceed the baseline T2I model on both datasets and all metrics. Ag3 (the LLM agent that does not explicitly utilize the belief graph) show superior performance across all metrics.

The plots in Figure 3 show the T2T, I2I, T2I and NLL metrics, averaged across all images in the ImageInWords dataset, per turn for 15 turns. We see that the multi-turn agents all improve in every metric as they increase the number of interactions. Interestingly we see the T2T and the T2I VQA similarity metric seems to plateau or decrease after 10 interactions, while the I2I scores continue to

---

[5]Text embeddings are obtained from Embeddings API: https://ai.google.dev/gemini-api/docs/embeddings.

[6]This approximation does not account for potential similarities in the names of entities or attributes. This could lead to approximation errors if, for example, the model confuses "Persian cat" with "Siamese cat" due to their similar names. Addressing this limitation would require incorporating semantic similarity measures into the NLL computation.

increase. The NLL metric shows large performance gains of the Ag3 agent in comparison to all other methods. The plots in Figure 10 shows the T2T DSG metrics.

| Dataset | Model | T2T ↑ | I2I (DINO) ↑ | T2I (VQAScore)↑ | NLL↓ | DSG (T2T)↑ |
|---|---|---|---|---|---|---|
| Coco-Captions | T2I | 0.8757±.03 | 0.5170±.16 | 0.2976±.45 | 520.0645±161.3 | 0.5904±.05 |
| | Ag1 | 0.9440±.02 | 0.6269±.12 | 0.5831±.49 | 508.4014±158.5 | 0.7555±.08 |
| | Ag2 | 0.9461±.02 | 0.6141±.13 | 0.6632±.46 | 481.7224±154.5 | 0.8344±.08 |
| | Ag3 | **0.9501**±.02 | **0.6575**±.10 | **0.7751**±.39 | **446.5679**±151.8 | **0.9001**±.05 |
| ImageInWords | T2I | 0.8807±.02 | 0.5154±.15 | 0.3711±.47 | 459.9053±200.2 | 0.6815±.70 |
| | Ag1 | 0.9429±.02 | 0.5548±.15 | 0.5058±.48 | 449.8927±196.1 | 0.8162±.08 |
| | Ag2 | 0.9382±.02 | 0.5645±.15 | 0.5701±.48 | 444.5227±193.7 | 0.8791±.07 |
| | Ag3 | **0.9418**±.02 | **0.5875**±.14 | **0.6624**±.45 | **429.4636**±194.5 | **0.9124**±.06 |
| DesignBench | T2I | 0.8740±.02 | 0.5439±.12 | 0.3528±.48 | 320.8898±93.7 | 0.6074±.08 |
| | Ag1 | 0.9365±.02 | 0.5943±.12 | 0.6848±.46 | 295.1974±69.2 | 0.8285±.08 |
| | Ag2 | 0.9384±.02 | 0.6417±.11 | 0.8553±.34 | 271.2604±81.9 | 0.9181±.06 |
| | Ag3 | **0.9429**±.02 | **0.6924**±.12 | **0.9545**±.21 | **257.4352**±67.5 | **0.9485**±.04 |

Table 1: Automatic evaluation results on **Coco-Captions**, **ImageInWords**, and **DesignBench**. Agents show large performance gains in all metrics over a standard T2I model alone.

## 5.3 ANALYSIS OF QUANTITATIVE RESULTS

The evaluations on the COCO-captions, ImageIn-Words, DesignBench datasets show similar results and highlight the same patterns across the different agents.

**Multi-Turn Agents show clear advantage:** The immediate take away is the baseline which does not use multi-turn interaction and instead passes in the original prompt into the T2I model performs worse than the multi-turn agents on all metrics on both datasets. This confirms our hypothesis that the current T2I agents often produce less desirable images given ambiguity in prompts. In Figure 2 we see real outputs of the multi-turn set up with the Ag3 agent.

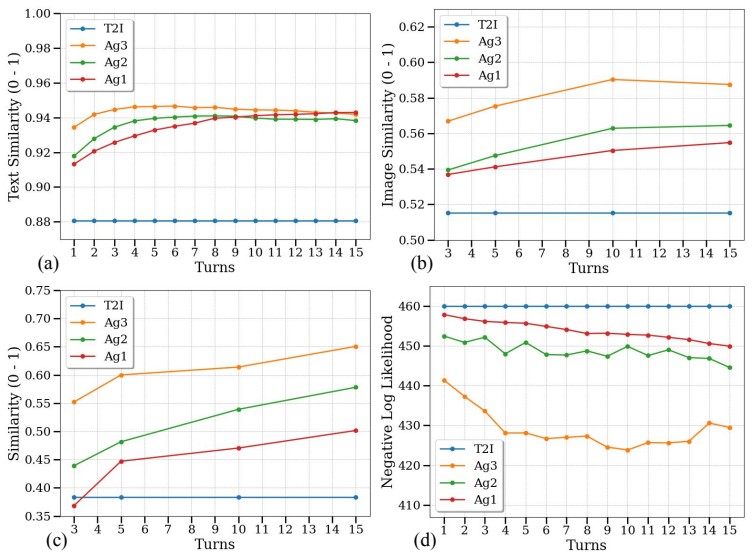

Figure 3: **ImageInWords** results, including (a) T2T, (b) I2I, (c) T2I, (d) NLL scores. Agents trend to increase performance up to 10 turns.

**LLMs being a part of agents play a significant role:** The best performers (Ag2 and Ag3) both query and LLM to provide a question to ask the user based on contextual information such as the belief graph and conversation history. They query the LLM to construct a concise and clear question but don't impose further constraints on the question construction. Ag1 provides a programatic template for how the LLM should construct the question based on its belief graph and does not provide any conversation history information. Examples of dialogs and the generated questions produced by the three agents can be found in the Appendix in Figure 4. This figure demonstrates that the templated question creation leads to extremely specific questions that often gather minimal information in return. This is an intrinsic limitation of hard coded question selection strategy but also can be an issue of the heuristic scores we defined for question selection in Ag1. In contrast, Ag2 and Ag3 generate questions that are more open-ended thus allowing the user to provide more nuanced details which in consequence enhance the Agent's image knowledge.

| Feature | V. Unlikely (%) | Unlikely (%) | Could Help (%) | Likely (%) | V. Likely (%) |
|---|---|---|---|---|---|
| Clarifications | 3.5 | 5.6 | 31.5 | 37.8 | 21.7 |
| Entity Graph | 4.2 | 7.7 | 35 | 32.9 | 20.3 |
| Relation Graph | 7 | 7 | 37.1 | 28.7 | 20.3 |

Table 2: Perceived helpfulness of proposed features (% of users) rated by 143 raters.

**Question prompts with question-asking principles show advantage over those with beliefs:** The Ag3 agent (which uses an LLM with question generation instructions about entity, attributes etc related to the belief) dominates across both datasets on every metric. Ag2 uses the belief explicitly to construct questions by passing the belief into the LLM as information from which to generate the next question. When inspecting the reasoning steps of Ag2, we found that Ag2 excessively relies on importance scores in beliefs to ask questions, and if the importance scores are not estimated properly, the quality of the questions decreases.

### 5.4 HUMAN STUDY

In order to get real user feedback, we performed a human survey with the objective of understanding user frustrations to validate whether our potential solutions could help with their use of T2I models. We gathered data from 143 participants who all identified to be regular T2I users (at least once a month). Participants were presented with four hypothesized frustrations (prompt misinterpretation, many iterations, inconsistent generations, incorrect assumptions) and three potential mitigating features (clarifications, entity graph, relationship graph; more details in Appendix §H).

As reported in Table 4 (in Appendix), the results confirmed the prevalence of hypothesized frustrations amongst users, with 83% experiencing occasional, frequent, or very frequent frustration due to prompt iterations, followed by 70% for misinterpretations, 71% for inconsistent generations, and 60% experiencing frustration due to incorrect assumptions. Most acutely 55% of participants reported frequent or very frequent frustration due to the prompt iteration frequency necessary. In Table 2, we report the mitigation features that are likely to help. Clarifications reported the highest likelihood to help current workflows (91% could / likely / very likely to be helpful), followed by entity graphs (88% could / likely / very likely to be helpful) and relationship graphs (86% could / likely / very likely to be helpful). Clarifications were expected to deliver value immediately / very soon by 58%.

Overall these suggest strong user desire for & likelihood for success of features that reduce iterations and mitigate misinterpretations in T2I generation. Full explanations of the hypothesized frustrations, mitigation and responses splits are in §H. All respondents were compensated for their time as per market rates, and were recruited by our vendor to ensure diversity across age, gender, and T2I usage in terms of models, frequency and purpose (work and non work).

### 6 DISCUSSION AND CONCLUSION

This work introduces a design for agents that assist users in generating images through an interactive process of question-asking and belief graph refinement. By dynamically updating its understanding of the user's intent, the agent facilitates a more collaborative and precise approach to image generation.

**Modular design.** Our agent prototypes are highly modular: the agents use frozen T2I models to generate images based on the prompts that the agent updated. Therefore when a better off-the-shelf T2I model becomes available, it can be directly plugged into the agents and the system will achieve better performance without any additional adaptation[7].

**Future work.** Alternative to the modular design, one can explore generating images directly from belief graphs and fine-tuning LLM/VLMs on text/image trajectories that include asking questions. These may require a) collecting data such as gold-standard trajectories or annotations on the quality of trajectories of human-agent conversations and b) new approaches to fine-tune the model on multi-turn trajectories of images and text, which can potentially improve the performance of the agent.

---

[7]T2T scores in Table 1 ablates the T2I model and only performs similarity on the captions. Our agents have achieved a 92%+ T2T score, showing that their performance can be boosted by adopting better T2I models.

ETHICS STATEMENT

Our proposed T2I agents are equipped with better tools (belief graphs) for interpretability and controllability. Presenting the agent's belief graph can be a generalizable method for AI transparancy, which is an important factor given the increasing complexity of modern AI models.

By asking clarification questions, our proposed agents may enable a more customizable and personalized content creation experience. Because different groups of people may perceive harmfulness of contents differently, learning more about the user through clarification questions can potentially mitigate risks of generating contents that can be offensive to each specific user.

REPRODUCIBILITY

We plan to release all code and DesignBench upon publication. All implementation details and prompts used in this work can be found in the appendix. All models we used in this work are publicly accessible with APIs linked in the experiments.

ACKNOWLEDGEMENTS

We would like to thank Jason Baldridge and Zoubin Ghahramani for insightful discussions on multi-turn T2I and belief states, Mahima Pushkarna for the help and consultation on user study. We would also like to thank Richard Song and Noah Fiedel for feedback on the paper.

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

## A  NOVELTY AND CONTRIBUTIONS

In this section, we emphasize the novelty and contributions of this work.

1. System design of proactive T2I agents:
   - Novel human-agent interaction modalities: Prior to our work, human users typically interact with current T2I systems by giving additional instructions or refining the prompt. To the best of our knowledge, our work is the first to propose a proactive T2I agent system that is able to ask clarification questions and present its belief graph for the user to edit.
   - Novel human-agent interaction interface: We designed a new interface to best enable the clarification and belief graph interaction modalities. We have not seen these features in any T2I, or other generative media apps that are publicly live to date, signifying to us total uniqueness. Our human studies showed that at least 85
   - Novel design of different T2I agents that enable the proposed interaction modalities. Please see Section 4 of the paper for the full details of the design principles and construction of those T2I agent prototypes (Ag1, Ag2, Ag3).

2. Our belief graph significantly differs from the classic belief state in the following ways:
   - Hardcoded predicates v.s. Automatically-generated predicates: Traditionally, constructing classic symbolic belief states requires a pre-defined set of predicates such as "on(a, b)", "is_red(a)", "at_position(robot, x, y, z)" and it is non-trivial to learn new predicates that can be used and generalized to new tasks (Pasula et al., 2007; Xia et al., 2019). Typically the pre-defined set of predicates are written by system developers and hardcoded into classic AI systems (Fikes & Nilsson, 1971).
   - Our belief graph does not require any pre-defined set of predicates. Instead, we propose to construct symbolic beliefs using a sequential in-context learning (ICL) method with LLMs. This method first generates a list of entities together with their descriptions conditioned on the user description of image; then, we add each entity to the context, and let the LLM generate a list of attributes and values (this step is done in parallel across entities); and finally, we add all entities to the context and let the LLM generate relations and their attributes. Our method can be generalized across a wide range of T2I tasks and achieve high performance (see our comprehensive results on Coco, Imageinwords, DesignBench). We have included a pseudo code for this method in the paper.
   - Application to planning v.s. T2I: To the best of our knowledge, classic symbolic belief states are mostly used for robot planning, and we are the first to use symbolic beliefs to assist T2I tasks. Data structure for symbolic states / belief states – Set v.s. Graph: Because of the application to planning, a symbolic world state is usually implemented and stored as a set or list of literals (atoms or negation of atoms where atoms are instantiated predicates (Alkhazraji et al., 2020; Garrett et al., 2020a;b) so that whenever an action is applied, the agent can apply transition by adding and deleting items in the set according to the precondition and effect of the action.
   - For T2I tasks, it is more convenient to use a graph to represent the world state associated with an image, since entities and relations naturally form a set of nodes (entities) and edges (relations between entities). Each component of the graph can also have probabilities, making it easy to turn a world state into a belief state using the same data structure. Hence we represent T2I agent beliefs using graphs. The agent can directly update the graph for each transition instead of going through a set or list.
   - Interpretability and controllability: The graph structure makes our agent belief more interpretable than traditional belief states, since we can visualize and progressively disclose the graph to the human user. Moreover, each node or relation in the belief graph has associated descriptions, making it easy for the user to understand and potentially edit every component of the belief graph. In our human studies, about 85% of raters found the belief graph useful. To the best of our knowledge, our work is the first to use the graph-based belief state for human-AI interaction.

3. Automated evaluation of T2I agents: We propose a novel automated evaluation approach for T2I agents using self-play. The agent interacts with a simulated user that has access to the original image and its long caption. See Section 5.1 (and C.4) for the full details of how the

simulated user is constructed. This evaluation pipeline is easy to use and can help the future development of T2I agents.

4. DesignBench: We envision that a significant fraction of T2I users are artists and designers, and it is important to ensure that T2I agents are evaluated for these use cases. Hence we create DesignBench, featuring photo-realism, animation and multiple styles with short and long captions. DesignBench can be directly plugged into our automated evaluation to streamline the evaluation process.

## B  FORMALISM OF THE AGENT AND ITS OBJECTIVE

We define an interactive T2I agent as a $\langle B, A, O, \tau, \pi \rangle$ tuple, where we have

- $S$: a representation space of images,
- $B$: a space of agent beliefs,
- $A$: a space of actions that the agent can take,
- $O$: a space of agent observations of the user,
- transition function $\tau : B \times A \times O \mapsto B$ for updating beliefs given new interactions,
- action selection strategy $\pi : B \mapsto A$, which specifies which action to take given a belief.

For each user-initiated interaction, we assume that there exists a specific intent $s \in S$, where $S$ is the space of all possible user intents. For a T2I task, we assume that the intent is the image the user would like to generate, and the intent stays the same throughout the interaction with an agent. We discuss more about the validity of this assumption in §6.

Each type of T2I agents can have a unique user intent representation, belief representation, construction of the action space, and user interface design to obtain observations of users.

In §4, we show the examples for these components.

We use a score function, $f : B \times S \mapsto \mathbb{R}$, to evaluate the alignment between an agent belief and a user intent at any turn of the interaction. Function $f$ can only be evaluated in hindsight once the user intent is revealed. The agent does not have direct access to function $f$ since the user intent is hidden from the agent. However, the agent may construct a probabilistic distribution over function $f$ based on its belief about the user intent. The goal of the agent is to maximize function $f$ with as few turns of interaction with the user as possible.

## C  VISUALIZATION OF MULTI-TURN AGENT-USER DIALOGS AND GENERATED IMAGES

In Figure 4, we show examples of multi-turn dialogs between simulated users and the three agents in Section 5. We also visualize the generated images in Figure 5, Figure 6, Figure 7 and Figure 8.

## D  IMPLEMENTATION DETAILS (FOR ALL AGENTS IN OUR EXPERIMENT)

We propose three distinct T2I agents, each characterized by a unique configuration of $\langle B, A, O, \tau, \pi \rangle$ tuples:

- *Ag1: Heuristic Score Agent*: this agent incorporates a human-defined heuristic score based on the belief to guide question generation. This heuristic score reflects the perceived importance of different aspects of the belief in driving the conversation forward;
- *Ag2: Belief-prompted Agent*: This agent leverages an LLM to generate questions by processing both the conversation history and a structured representation of the belief.
- *Ag3: Principle-prompted Agent*: This agent generates questions directly from the conversation history, relying solely on the implicit knowledge and reasoning capabilities of the underlying Large Language Model (LLM). It does not employ an explicit, structured belief representation;

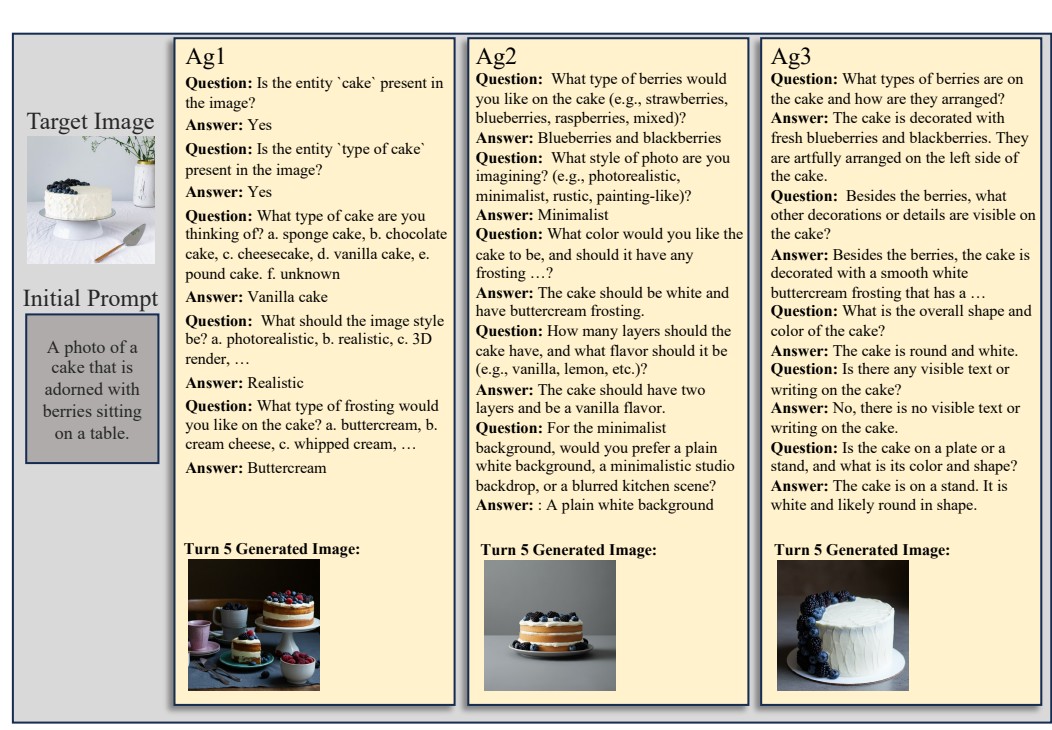

Figure 4: Real multi-turn dialogs generated by the Ag1, Ag2, and Ag3 agents on an image from DesignBench. The figure additionally shows the image generated after the 5 turn dialog per agent.

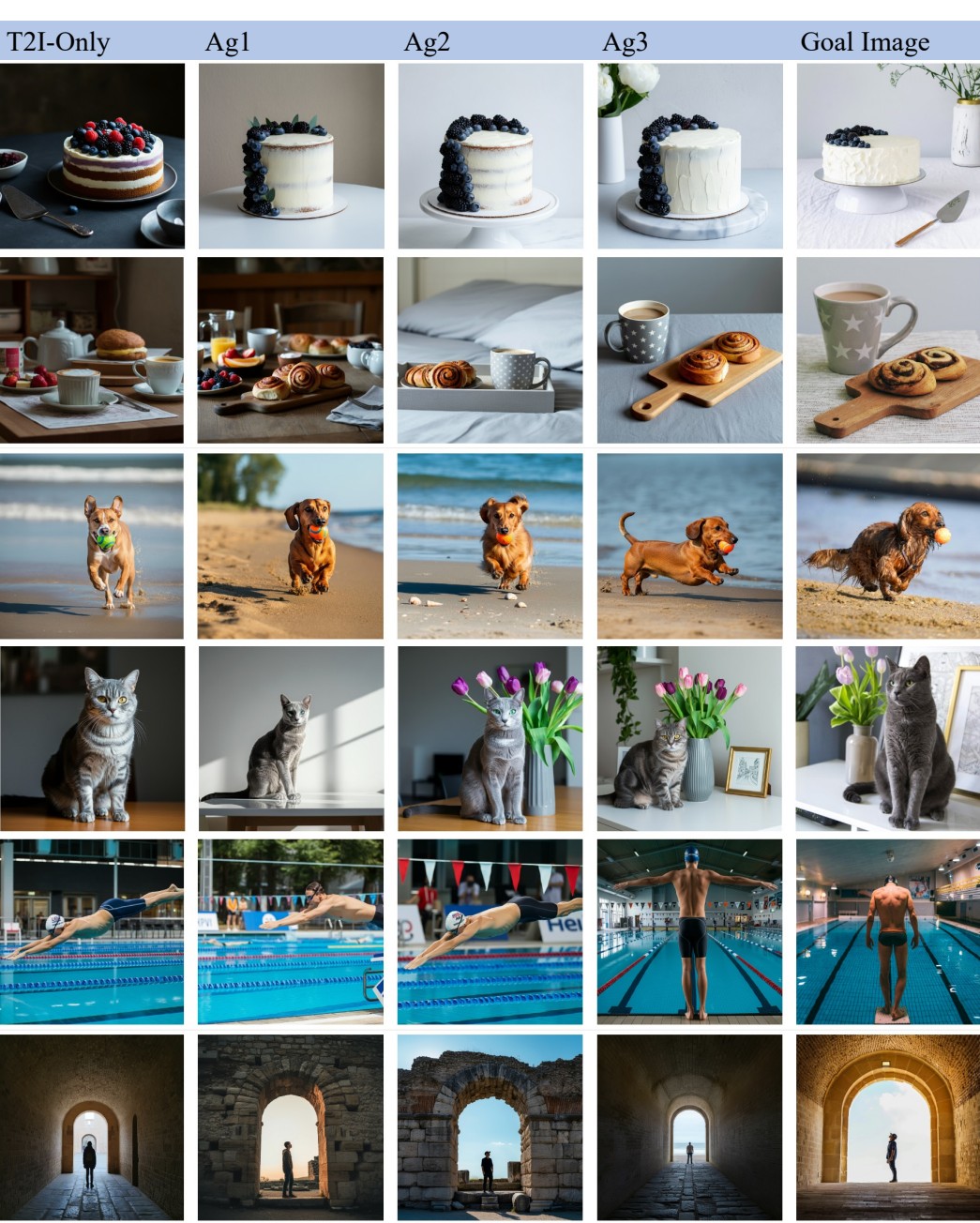

Figure 5: Agent Generated Image Outputs on DesignBench: a chart of the generated image outputs of the four main Agent types in comparison to the goal image. Each column displays the output of a different agent and the right most column shows the goal image that the agents aimed to recreate. Each agent was provided with the same starting prompt and iterated for 15 turns, with the exception of the "T2I" agent column which produces an image from the starting prompt. Ag1, Ag2 and Ag3 refer to the Agents described in §D. Each agent uses the same T2I model to produce the final image. The goal images displayed here are from our DesignBench dataset described in the experiments section.

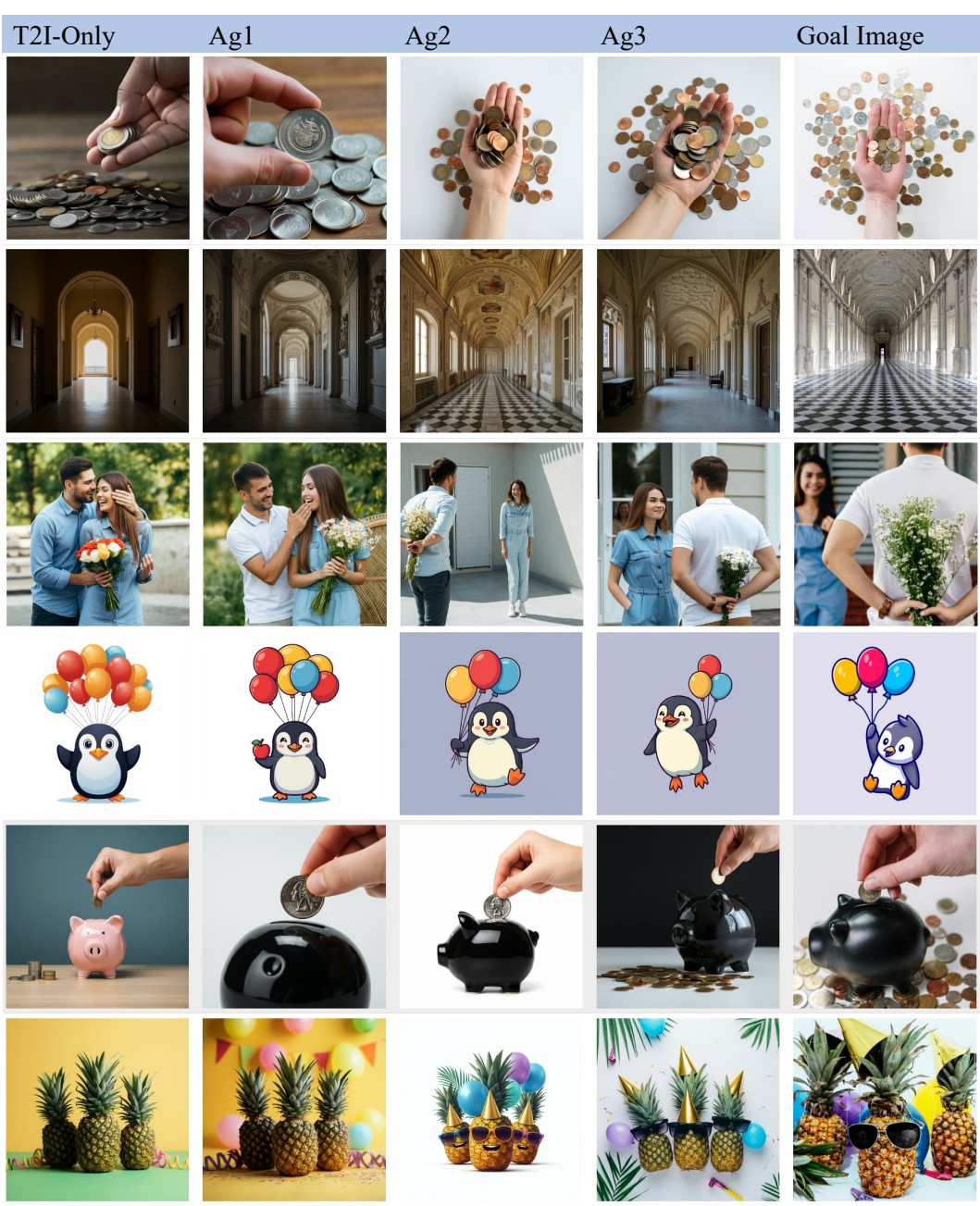

Figure 6: Agent Generated Image Outputs on DesignBench (Continued): a chart of the generated image outputs of the four main Agent types in comparison to the goal image. Each column displays the output of a different agent and the right most column shows the goal image that the agents aimed to recreate. Each agent was provided with the same starting prompt and iterated for 15 turns, with the exception of the "T2I" agent column which produces an image from the starting prompt. Ag1, Ag2 and Ag3 refer to the Agents described in §D. Each agent uses the same T2I model to produce the final image. The goal images displayed here are from the DesignBench dataset described in the experiments section.

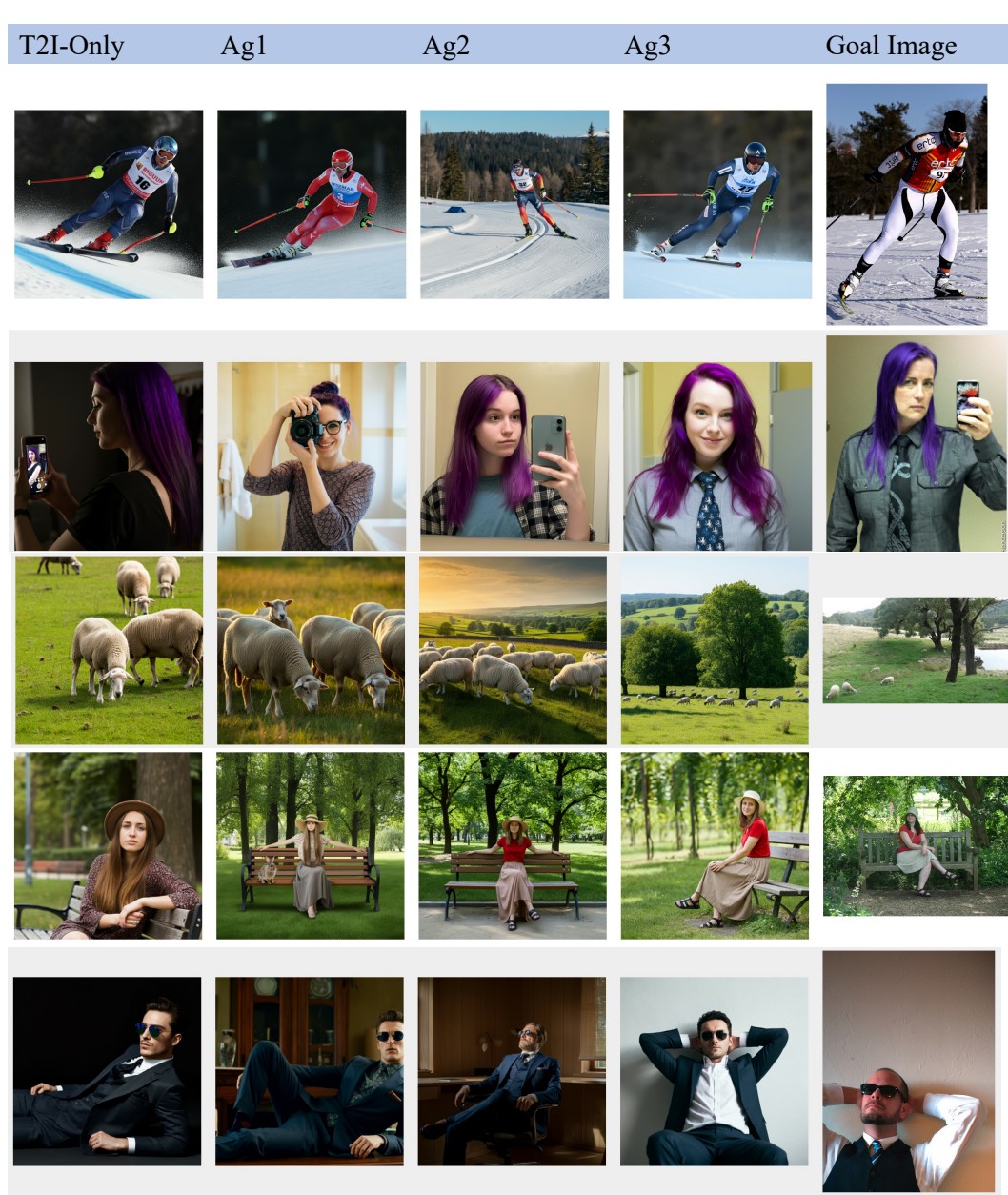

Figure 7: Agent Generated Image Outputs (Coco-Captions Validation): a chart of the generated image outputs of the four main Agent types in comparison to the goal image. Each column displays the output of a different agent and the right most column shows the goal image that the agents aimed to recreate. Each agent was provided with the same starting prompt and iterated for 15 turns, with the exception of the "T2I" agent column which produces an image from the starting prompt. Ag1, Ag2 and Ag3 refer to the Agents described in §D. Each agent uses the same T2I model to produce the final image. The goal images displayed here are from the Coco-Captions Chen et al. (2015) dataset described in the experiments section.

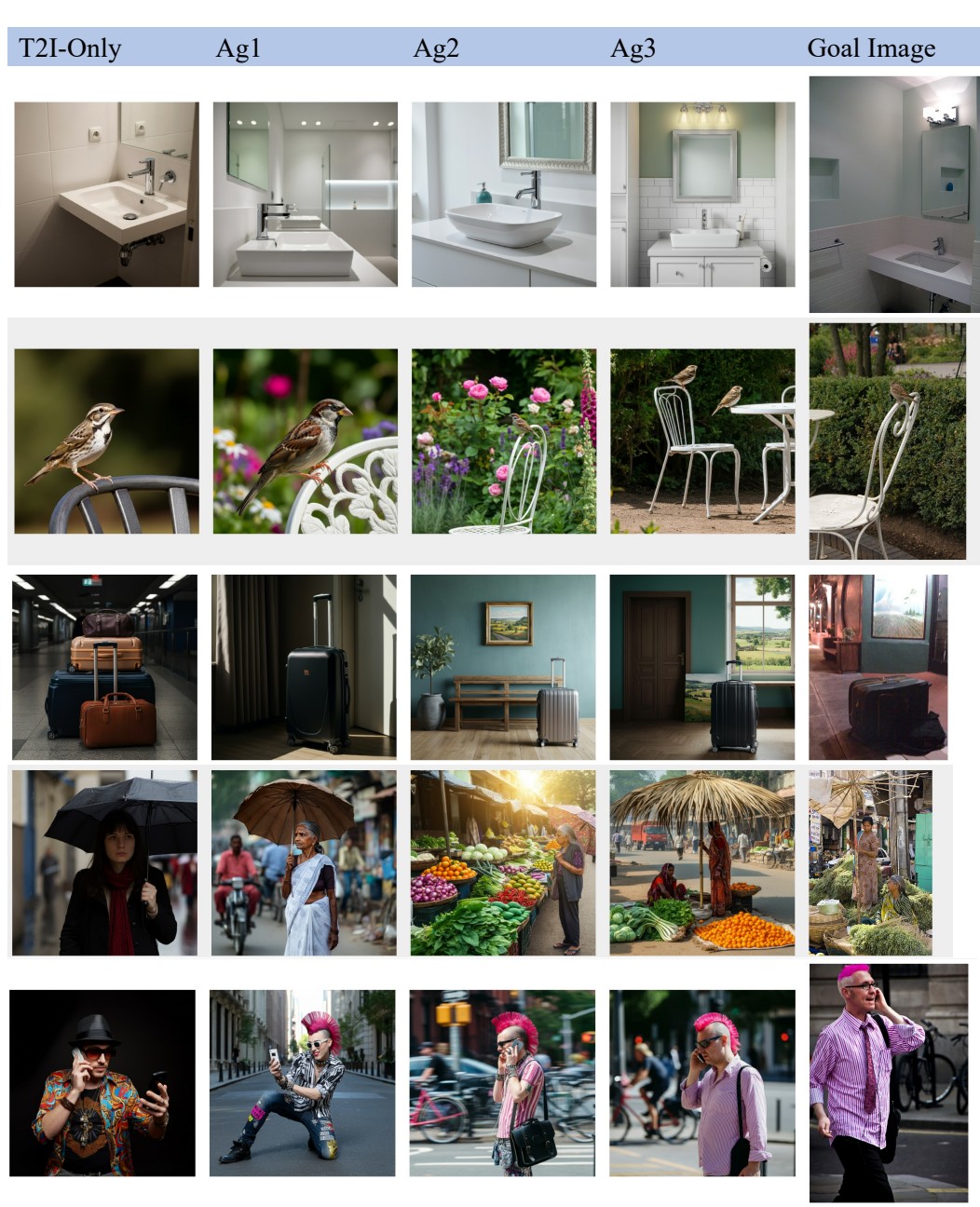

Figure 8: Agent Generated Image Outputs (Coco-Captions Validation): a chart of the generated image outputs of the four main Agent types in comparison to the goal image. Each column displays the output of a different agent and the right most column shows the goal image that the agents aimed to recreate. Each agent was provided with the same starting prompt and iterated for 15 turns, with the exception of the "T2I" agent column which produces an image from the starting prompt. Ag1, Ag2 and Ag3 refer to the Agent described in the methods section. Each agent uses the same T2I model to produce the final image. The goal images displayed here are from the Coco-Captions Chen et al. (2015) dataset described in the experiments section.

### D.1 IMPLEMENTATION OF STATE

Our agent's state is represented in two complementary forms: (i) Merged prompt: This is a natural language representation that summarizes the entire conversation history up to the current turn. It provides a comprehensive textual overview of the user's requests, feedback, and any clarifications exchanged with the agent. (ii) Belief: This is a symbolic representation derived from the merged prompt. It parses the natural language text into a structured format, capturing key elements like entities, attributes, relationships, and associated probabilities. This structured representation facilitates more precise reasoning and decision-making by the agent.

**Prompt Merging.**   An LLM (§D.11) summarizes the latest interaction, encapsulating the agent's question and the user's response into a concise textual representation. This step distills the essential information exchanged during the interaction. Another LLM (§D.12) merges the summarized interaction with the existing merged prompt, which contains the accumulated information from previous interactions. This creates an updated prompt that reflects the evolving understanding of the user's intent.

**Belief Parsing.**   See an example of the belief state fig. 9. We employ three specialized parsers trained via in-context learning (ICL): entity parser (§D.8) analyzes the user prompt to identify and extract a list of relevant entities.; attribute parser (§D.9) takes user prompt and an entity as the input to extract a list of attributes associated with that entity; relation parser (§D.10) takes the user prompt and a list of entities as input and identifies relationships between those entities. Each entity is associated with meta information like name, importance to ask score, description, probability of appearing, a list of attributes like color, position, etc [8]. Each attribute contains meta information like name, importance to ask score, a list of possible values for the attribute along with their associated probabilities, etc. Each relation includes meta information such as: name, description, spatial relation, importance to ask score, entity 1 and entity 2, whether the relation is bidirectional, etc.

---

[8]**Name** is a unique identifier for the entity; **Importance to ask score**: A numerical value indicating the entity's perceived importance in satisfying the user's request. Entities with higher scores are prioritized during question generation, as they are likely to reduce uncertainty and contribute significantly to the final image; **Description** provides a textual description of the entity; **probability of appearing** estimates likelihood of the entity being present in the generated image; **Attributes** is for understanding the detailed attributes of the entities.

Belief state

Entities

Rabbit
Attribute Name: color, importance_score: 0.9,
          candidates: [brown: 0.25, white: 0.25, grey: 0.2, black: 0.15, …]
Attribute Name: breed, importance_score: 0.3,
          candidates: [Dutch: 0.2, Mini Lop: 0.15, Netherland Dwarf: 0.15, …]
Attribute Name: expression, importance_score: 0.5,
          candidates: [scared: 0.8, determined: 0.1, playful: 0.1]
                                    ……

Dog
Attribute Name: breed, importance_score: 0.8,
          candidates: [Labrador Retriever: 0.15, Golden Retriever: 0.15, German
          Shepherd: 0.15, Bulldog: 0.1, Beagle: 0.1, …]
Attribute Name: coat_color, importance_score: 0.7,
          candidates: [brown: 0.2, black: 0.2, white: 0.2, …]
Attribute Name: coat_style, importance_score: 0.6,
          candidates: [long: 0.3, short: 0.3, fluffy: 0.2, shaggy: 0.1, wavy: 0.1]
Attribute Name: hat, importance_score: 0.5,
          candidates: [baseball cap: 0.2, bowler hat: 0.2, top hat: 0.2, …]
                                    ……
                                ……

Relations

Dog-Rabbit
importance_score: 0.9,
spatial_relation: [chasing: 1.0]

Coat-Dog
importance_score: 0.8,
adornment_relation: [wearing: 1.0]
                          ……

Figure 9: Agent Belief State: and example of the belief state for a given prompt in Figure 1.

We employ the merged prompt across all agent variations (*Ag1, Ag2, Ag3*) to generate images at each turn of the interaction. Belief is being used in *Ag1* and *Ag2*, which utilize belief parsing to extract structured representations from the user's input. *Ag3* relies solely on the LLM's inherent ability to grasp the user's needs from the conversation history and merged prompt, without explicit belief state construction.

## D.2 Implementation of Action

From an information theoretic perspective, an optimal action is the one that maximizes the information gain between the observation and the belief, i.e. $a_t = \arg\max_a H(o_{i-1}; b_{i-1} \mid a) - H(o_i; b_i \mid a)$. However, directly optimizing this objective can be computationally challenging. Therefore, we explore several heuristic strategies to effectively reduce uncertainty:

- Maximize the overall heuristic importance score ($MHIS$):This strategy focuses on maximizing the overall importance score of the entities, attributes, and relations within the belief. We further ask a question regarding an attribute or relation by maximizing the overall heuristic importance score. The score can be modeled as:

$$max_{e,a,c,r}(IS(\text{e}) * IS(\text{a}) * P(\text{e}) * Ent(\text{c}), IS(\text{r}) * P(\text{r}) * Ent(\text{c})) \qquad (1)$$

  Here $IS, P, Ent$ represents importance to ask score, probability of appearing, and entropy of the probabilities respectively and $e, a, c, r$ represents entity, attribute, candidate list, relation respectively.
- Ask Important Clarification Question based on belief ($AICQ_B$): This strategy leverages the structured information within the belief. We provide the LLM with the user prompt, conversation history, and the current belief, utilizing an ICL prompt (§D.14) to guide question generation. The LLM then formulates a clarification question aimed at eliciting information about key features of the image, naturally prioritizing those with higher *Importance to ask score* within the belief.
- Ask Important Clarification Question directly ($AICQ_{base}$): This strategy relies on the LLM's inherent ability to identify important aspects of the user prompt and conversation history. The LLM (§D.13) generates an important clarification question based on its implicit understanding of the user's needs, without explicitly relying on the structured information in the belief.

*Ag1* employs $MHIS$ strategy for question generation. This strategy leverages the importance scores assigned to entities, attributes, and relations within the belief state. It identifies the element with the highest heuristic importance score and formulates a question aimed at eliciting further information about that specific element. The question is then verbalized using the LLM described in Section §D.15.

*Ag2* utilizes the parsed belief state as the basis for question generation. It employs the $AICQ_B$ strategy, which leverages the structured information within the belief state to generate targeted clarification questions.

*Ag3* relies solely on the conversation history for question generation. It employs the $AICQ_{base}$ strategy, which leverages the LLM's ability to understand the ongoing dialogue and identify key areas requiring further clarification.

## D.3 Implementation of Transition

**Belief Updating.** Both *Ag1* and *Ag2* require belief updating to incorporate new information gained during the interaction. At each turn, we perform prompt merging to create a comprehensive prompt that summarizes the conversation history. This merged prompt is then used for belief parsing to obtain an updated belief state. For *Ag2*, this updated belief state directly informs the subsequent interaction. For *Ag1*, it incorporates additional post-processing mechanisms to enhance memory and prevent redundant questioning: (i) Redundancy elimination: If an attribute or relation has already been addressed in the conversation history, the corresponding user response is assigned as the sole candidate with a probability of 1.0, and its importance score is set to 0. This prevents the agent from repeatedly asking about the same information. (ii) Information retention: If an attribute or relation from the conversation history is absent in the updated belief state, it is explicitly added. This ensures

that the agent retains crucial information even if it's not explicitly present in the latest parsed belief state.

### D.4    USER SIMULATION

To simulate end-to-end agent-user interactions, we implement a user simulator that mimics human question-answering behavior. This simulator operates as follows:

- It generates a belief state based on a ground truth prompt, representing the user's intended image. This serves as the simulator's internal representation of the desired image.
- Mirroring the $AICQ_B$ strategy, the simulator takes the ground truth prompt, conversation history, and its current belief state as input. It then leverages an ICL prompt (see §D.14) to generate a response to the agent's question. This ensures that the simulator's answers are consistent with its internal belief state and the ongoing conversation.

### D.5    AG1: HEURISTIC SCORE AGENT

The *Heuristic score agent* leverages the importance scores and probabilities within the belief state to guide its question-asking strategy. The underlying principle is to identify and inquire about the entity, attribute, or relation that exhibits both high importance and high uncertainty. This aligns with the uncertainty reduction principle discussed in §4.1.1, which emphasizes minimizing uncertainty through targeted questioning. To achieve this, we define a *heuristic importance score* as formulated in Equation 1, and the agent then selects the attribute or relation with the highest heuristic importance score as the focus of its inquiry. To facilitate easy answering, we utilize an LLM to generate user-friendly questions with multiple-choice options. For example, the agent might ask: *What color of the rabbit do you have in mind? a. black , b. white, c. brown. d. unkown. If none of these options , what color of the rabbit do you have in mind?*. This format allows users to simply select the most appropriate option or provide their own answer if needed.

Here's a summary of Ag1's implementation: (i) **State Representation**: The agent's state comprises the merged prompt and the current belief state. (ii) **Select Action**: $MHIS$ strategy is employed to identify the attribute or relation of interest based on the heuristic importance score. (iii) **Verbalize Action**: An LLM (§D.15) is used to generate a clear and concise question about the selected attribute or relation. (iv) **Answer Question**: The user simulator provides an answer to the agent's question, mimicking human response behavior. (v) **Transition**: The agent updates the merged prompt with the new information, re-generates the belief state based on the updated prompt, and applies the post-processing logic outlined in §D.3 to ensure consistency and prevent redundancy.

### D.6    AG2: BELIEF-PROMPTED AGENT

The *Ag2* agent incorporates the belief state into its decision-making process but adopts a different approach compared to *Ag1*. Instead of relying on a heuristic score, *Ag2* leverages the full capacity of an LLM to generate questions. It provides the LLM with comprehensive information, including the merged prompt, belief state, and conversation history, allowing the LLM to formulate the most informative questions possible. To guide the LLM towards generating effective questions, we incorporate specific instructions in the prompt, emphasizing the following principles: *The question should be as concise and direct as possible. The question should aim to obtain the most information about the style, entities, attributes, spatial layout and other contents of the image. Remember to ask for information that are critical to knowing the critical details of the image that is important to the user. The question should reduce your uncertainty about the user intent as much as possible.*

Here's a summary of Ag2's implementation: (i) **State Representation**: The same as *Ag1*, the agent's state consists of the merged prompt and the current belief state. (ii) **Select Action**: $AICQ_B$ strategy is employed, which leverages an LLM to generate a question based on the comprehensive input information. (iii) **Verbalize Action**: Since the LLM directly generates the question, no separate verbalization step is required. (iv) **Answer Question**: The user simulator provides an answer to the agent's question. (v) **Transition**: The agent updates the merged prompt with the new information and re-generates the belief state based on the updated prompt.

## D.7 AG3: PRINCIPLE-PROMPTED AGENT

A simple and effective implementation of LLM-based multi-modal dialogue systems is to use the context to store the history of conversations between the system and the user, and directly generate the next response based on the context.

To align with the principles outlined in §4.1.1, we guide the LLM's question generation with a prompt (§D.13) that emphasizes all principles: *Based on the original prompt and chat history please provide a question to ask about the image. The question should be as concise and direct as possible. The question should aim to learn more about the attributes and contents of the image, the objects, the spatial layout, and the style.*. The prompt also includes the history of conversation. This strategy aims to generate questions that are easy for users to understand and answer, while effectively reducing the agent's uncertainty about the desired image.

Here's a summary of Ag3's implementation: (i) **State Representation**: The same as *Ag1* and *Ag2*, the agent's state consists of the merged prompt and the current belief state. (ii) **Select Action**: $AICQ_{base}$ strategy is employed, which leverages an LLM to generate a question based on the conversation history. (iii) **Verbalize Action**: The LLM directly generates the question, so no separate verbalization step is needed. (iv) **Answer Question**: The user simulator provides an answer to the agent's question. (v) **Transition**: The same as *Ag2*, the agent updates the merged prompt with the new information and re-generates the belief state based on the updated prompt.

## D.8 ENTITY PARSER PROMPT INSTRUCTION

```
1   Given a text-to-image prompt list out all the entities that are mentioned in the prompt.
2
3   **Explicit Entities:** List all clearly stated entities within the prompt (people, objects, animals, locations, etc.).
4   **Implicit Entities:** Identify potential entities that are implied or strongly suggested by the prompt, even if not explicitly mentioned.
5   **Background Entities:** Deduce relevant background elements which could impact the image generation from the prompt or context, including:
6       **Weather:** If the scene or mood suggests specific weather conditions (sunny, rainy, stormy, etc.).
7       **Location:** If a general or specific setting is hinted at (indoors, outdoors, a particular city/landscape, etc.).
8       **Time of Day:** If the prompt implies a certain time (dawn, midday, dusk, night).
9       **Mood or Atmosphere:** If the prompt evokes a particular emotion or ambiance (joyful, mysterious, peaceful, etc.).
10
11
12  The output should be list and each entry should be formated as a JSON dict with the following fields:
13
14  "name": The name of the entity.
15  "importance_to_ask_score": The importance score of asking a question about this entity to reduce the uncertainty of what the image is given the
          user prompt. Make sure that this is a number between 0 and 1, higher means more important. Consider these factors when assigning scores: 1.
          Increate the score for entities that are the primary focus or subject of the prompt; 2. increase the score for entities that could
          strongly influence the layout of the image, such as the position or portrayal of other entities in the scene; 3. significantlydescrease the
          score for entities that are already well specified in the prompt; 4. significantlyincrease the score for implicit entities that are likely
          to appear in the image and their appearance can significantly impact the image.
16  "description": A short description of the entity.
17  "entity_type": The type of this entitiy. It could be either explicit, implicit, background. No other value is allowed.
18  "probability_of_appearing": The probability of the entity appearing in the image. This is a number between 0 and 1. You should assign a probability
          with the following rules in mind:
19    1. If the prompt says an entity does not exist, assign a 0.0 probability. Because the entity does not exist, you should also assign 0 to
          importance_to_ask_score of this entity.
20    2. If the prompt indicates an entity definitly exists in the image, assign a 1.0 probability.
21    3. If the prompt does not say anything about the existence of the entity, assign a probability between 0 and 1. This probability is higher if the
          entity is more likely to appear in the image given the context specified by the prompt.
22    4. If the prompt says an entity exists but there is an indication that the entity is not likely to appear in the image, assign a probability
          between 0 and 1, higher if the entity is more likely to appear in the image.
23
24  Below is an example input and output pair:
25  Example1:
26  Input: {{
27    "user_prompt": "generate an image of a lionhead rabbit running on grass with sun shining. There is no trees in the background."
28  }}
29  Output: [
30      {{
31        "name": "rabbit",
32        "importance_to_ask_score": 0.5,
33        "description": "a lionhead rabbit",
34        "entity_type": "explicit",
35        "probability_of_appearing": 1.0
36      }},
37      {{
38        "name": "grass",
39        "importance_to_ask_score": 0.5,
40        "description": "grass",
41        "entity_type": "explicit",
42        "probability_of_appearing": 1.0
43      }},
44      {{
45        "name": "sun",
46        "importance_to_ask_score": 0.1,
47        "description": "sun is shining",
48        "entity_type": "explicit",
49        "probability_of_appearing": 0.3
50      }},
51      {{
52        "name": "sun light",
53        "importance_to_ask_score": 0.1,
54        "description": "sun light shining on the grass and the rabbit",
55        "entity_type": "explicit",
56        "probability_of_appearing": 1.0
57      }},
58      {{
59        "name": "tree",
60        "importance_to_ask_score": 0,
61        "description": "trees in the background",
62        "entity_type": "explicit",
63        "probability_of_appearing": 0
64      }}
65      {{
66        "name": "camera angle",
67        "importance_to_ask_score": 0.8,
68        "description": "the camera angle of the image",
69        "entity_type": "background",
70        "probability_of_appearing": 1.0
71      }},
72      {{
73        "name": "weather",
74        "importance_to_ask_score": 0.8,
75        "description": "weather",
76        "entity_type": "background",
77        "probability_of_appearing": 1.0
78      }},
79      {{
80        "name": "image style",
81        "importance_to_ask_score": 1.0,
```

```
82       "description ": "the  style  of the  image",
83       " entity_type ": "background",
84       " probability_of_appearing ":  1.0
85     }},
86     {{
87      "name": "background color ",
88      " importance_to_ask_score ":  0.8,
89      " description ": "the  background color  of  the  image",
90      " entity_type ": "background",
91      " probability_of_appearing ":  0.5
92     }}
93  ]
94
95  ...  [[a  few  additional  examples]]  ...
96
97
98  Identify  the  entities  given  the  input given  below.  Strictly   stick  to the  format.
99  Input :  {{
100    "user_prompt": "{user_prompt}"
101  }}
102  Output:
```

## D.9 ATTRIBUTE PARSER PROMPT INSTRUCTION

```
1  Given a text-to-image prompt and a particular entity described in the prompt, and your goal is to identify a list possible attributes that could
        describe the particular entity. Output Requirements:
2
3  1. if this attribute has already existed as an entity in other existing entity list, then do not include it.
4  2. the attribute candidate could be a mixed of values like 'color A and color B'.
5  3. The output should be a json parse-able format:
6
7  name(str): The name of the attribute.
8  importance_to_ask_score (float): The importance score of asking a question about this attribute to reduce the uncertainty of what the image is
        given the user prompt. This is a number between 0 and 1, higher means more important. Consider these factors when assigning scores: 1.
        Increate the score for attributes that are the primary attributes of an important entity; 2. significantly increase the score for
        attributes that could strongly influence the generation or portrayal of OTHER attributes in the scene; 3. descrease the score for
        attributes that are already well specified in the prompt. For example, a breed of a dog would impact other attributes like color, size, etc
        . So the breed attribute should have a higher importance score than color, size, etc. Assign a much lower score if the attribute's value is
        already mentioned in the user prompt.
9  candidates (List of names and probabilities): List of possible values that the attribute can take. Make sure to generate atleast 5 or more possible
        values. These should be realistic for the given entity. For each attribute, returns the probability that the user wants this candidate
        based on the user prompt. If it's already mentioned by the user, only generate one candidate (the mentioned one) and assign 1.0 as the
        probability. The sum of probabilities over all candidates shall be 1. Also infer the probability based on the prompt. For example, for a
        dog with breed Samoyed, the color attribute has a very high probability of white.
10
11 Below are two examples of input and output pairs:
12
13 Example 1:
14 Input: {{
15    "user_prompt": "generate an image of a white rabbit running on grass",
16    "entity": "rabbit",
17    "other_existing_entities": "grass"
18 }}
19 Output: [
20     {{
21       "name": "color",
22       "importance_to_ask_score": 0.9,
23       "candidates": {{"white":1.0}}
24     }},
25     {{
26       "name": "breed",
27       "importance_to_ask_score": 1.0,
28       "candidates": {{"Dutch": 0.20,
29                      "Mini Lop": 0.15,
30                      "Netherland Dwarf": 0.15,
31                      "Lionhead": 0.10,
32                      "Flemish Giant": 0.10,
33                      "Mini Rex": 0.10,
34                      "English Angora": 0.08,
35                      "Mini Satin": 0.05,
36                      "Himalayan": 0.05,
37                      "Californian": 0.02}}
38     }},
39     {{
40       "name": "age",
41       "importance_to_ask_score": 0.1,
42       "candidates": {{"adult": 0.6,
43                      "baby": 0.2,
44                      "senior": 0.2}}
45     }}
46   ]
47
48 ... [[a few additional examples]] ...
49
50 Generate attributes given the input given below. Do not include other entities in the attributes. Strictly stick to the format.
51 Input: {{
52    "user_prompt": "{user_prompt}",
53    "entity": "{entity.name}",
54    "other_existing_entities": "{existing_entities}"
55 }}
56 Output:
```

## D.10 RELATION PARSE PROMPT INSTRUCTION

```
1  Given a text−to−image prompt and a list of entity described in the prompt, your goal is to identify a list of entity pairs that have relations
       between them. Ignore entity pairs without relations . The output should be a json parse−able format (No comma after the last element of the
       list ):
2
3  Input :
4  user_prompt: the prompt from the user .
5   entities : a list of entities mentioned in the user_prompt.
6
7  Output:
8  name (str ): The name of the relation . Use ' entity1−entity2 ' as the format.
9  description ( str ): A short description of the relation .
10  spatial_relation ( map from potential relation candidates to probability ): Possible spatial relations between the two entities . If a relation is
       mentioned in the user prompt, assign 1.0 as the probability . The sum of probabilities over all relation candidates shall be 1.
11 importance_to_ask_score ( float ): The importance score of asking a question regarding this relation to reduce entropy. This is a number between 0
       and 1, higher means more important. Assign a higher score if the two entities are very important , the relation between them is very unclear
       , and the relation is very important for the layout of the image.
12 name_entity_1 ( str ): The name of the first entity .
13 name_entity_2 ( str ): The name of the second entity .
14  is_bidirectional (bool): Whether the relation is bidirectional .
15
16 Below is an example input and output pair :
17 Example1:
18 Input : {{
19  "user_prompt": "generate an image of a lionhead rabbit sitting on grass , and a eagle is flying through the sky. There is a tree in the
       background.",
20  " entity ": [" rabbit ", "grass ", "eagle ", "tree "]
21 }}
22 Output: [
23    {{
24      "name": "rabbit−grass ",
25      "description ": "rabbit sitting on grass ",
26      " spatial_relation ": {{"above": 0.8, "below": 0.0, "in front of": 0.0, "behind": 0.0, " left of": 0.1, "right of": 0.1}},
27      " importance_to_ask_score ": 0.1,
28      "name_entity_1 ":" rabbit ",
29      "name_entity_2 ": "grass ",
30      " is_bidirectional ": true
31    }},
32    {{
33      "name": "eagle−grass",
34      "description ": "eagle is flying through the sky",
35      " spatial_relation ": {{"above": 1.0, "below": 0.0, "in front of": 0.0, "behind": 0.0," left of": 0.0, "right of": 0.0}},
36      " importance_to_ask_score ": 0.1,
37      "name_entity_1 ":" eagle ",
38      "name_entity_2 ": "grass ",
39      " is_bidirectional ": false
40    }},
41    {{
42      "name": "tree−grass ",
43      "description ": "",
44      " spatial_relation ": {{"above": 0.5, "below": 0.0, "in front of": 0.0, "behind": 0.0, " left of": 0.25, "right of": 0.25}},
45      " importance_to_ask_score ": 0.1,
46      "name_entity_1 ":" tree ",
47      "name_entity_2 ": "grass ",
48      " is_bidirectional ": false
49    }},
50
51    ... [[a few additional examples]] ...
52
53   ]
54
55  Identify relationships between entities given the input given below. Strictly stick to the format.
56 Input : {{
57    "user_prompt": "{user_prompt}",
58    " entity ": "{entity_names}"
59 }}
60 Output:
```

## D.11 VERBALIZATION PROMPT INSTRUCTION

```
1  The chat history is as follows :
2  question : {action. verbalized_action } and answer: {observation }.
3  Turn the question and action into a single declarative sentence that describes the answer − do not phrase it as a question . Example output: the
       firetruck in the image is red .
```

## D.12 MERGE PROMPT PROMPT INSTRUCTION

```
1  You are writing a prompt for a text−to−image model based on user feedback. The original prompt is {prompt}. The user has provided some additional
       information : { additional_info }. Please write a new prompt for the text−to−image model. The new prompt should be a meaningful sentence or a
       paragraph that combines the original prompt and the additional information . Do not add any new information that is not mentioned in the
       prompt or the additional information . Make sure the information in the original prompt is not changed. Make sure the additional information
       is included in the new prompt. Make sure the new prompt is a description of an image. If the additional information or the original prompt
       specifically says that a thing does not exist in the image, you should make sure the new prompt mentions that this thing does not exist in
       the image. DO NOT generate rationale or anything that is not part of a description of the image.
```

## D.13  $AICQ_{base}$ PROMPT INSTRUCTION

```
1   ...  [[ Instruction  for  the  first  question ]]  ...
2
3   The  original  prompt  was: { self . original_prompt } – Based on the  original  prompt please  provide a question to ask about the image. The question
        should be as concise and direct as possible . The question should aim to learn more about the  attributes  and contents of the image, the
        objects , the  spatial  layout , and the  style . Make sure that  you question the answer within <question> and </question> markers
4
5   ...  [[ Instruction  for  the  following  question ]]  ...
6
7   Based on the chat  history  please  provide a new question to ask about the image. the chat  history  is as follows and is enclosed in  <chat_history>
        and </chat_history> markers:{ self. chat_history } </chat_history> The question should be as concise and direct  as  possible . The question
        should aim to learn more about the  attributes  and contents of the image, the objects , the  spatial  layout , and the  style . Make sure that  you
        question the answer within <question> and </question> markers.'
```

## D.14  $AICQ_B$ PROMPT INSTRUCTION

```
1   You are an intelligent  agent  that helps users generate images. Before  generating the image requested by the user , you should ask the most important
        clarification   questions to make sure you understand the key features of the image.
2   The user  describes  the image as : {user_prompt}.
3   The following is your  belief  of what the image contains , including the  entities ,  attributes  of each entity and relations between entities .
4   Each entity has "name"," descriptions ", "importance to ask score" and  " probability of appearing". "Name" is the  identifier  of the entity . "
        Descriptions " is  the  description  of the entity . "Importance to ask score" is how important it is for the agent to ask whether the entity
        exists . Probability of appearing" is the  probability  the agent estimated that  this  entity  exits in the image.
5
6   Each entity  has a list of  attributes . Each attribute  has "name", "importance to ask score" and "candidates ". "Name" is the  identifier  of the
        attribute . "Importance to ask score" is how important it is to ask about the exact value for the  attribute  of the entity . "Candidates" is a
        list of possible  values for the  attribute .
7
8   Each candidate  value has a probability  that describes how likely  this  candidate value should be assigned to the  attribute .
9   For example, " Attribute  Name: color, Importance to ask Score: 0.9, Candidates: [white: 0.5, black: 0.5]" means the color is  either white or black,
        each with 0.5  probability . If you ask about  attributes , you should ask about the  attribute  with the highest  uncertainty . Your uncertainty
        can be judged by the  probabilities . If the  probabilities  are 0.5 and 0.5, you are uncertain . If the  probabilities  are 0.1 and 0.9, you are
        fairly  certain .
10
11  The agent  belief  is :
12  { belief_state . __str__ () }
13
14  Based on the user prompt "{user_prompt}" and the  belief  of the agent , please provide a question to ask about the image. The question should be as
        concise and direct as possible . The question should aim to obtain the most information about the style , entities ,  attributes , spatial
        layout and other contents of the image. Remember to ask for information that are  critical  to knowing the critical   details  of the image that
        is important to the user . The question should reduce your  uncertainty  about the user  intent as much as possible . DO NOT ask question that
        can be answered by common sense. DO NOT ask question that are  obvious  to answer based on  the  user prompt "{user_prompt}". DO NOT ask any
        question about information  present in the following user–agent dialogue within <dialogue> and </dialogue> markers.
15
16  <dialogue>
17  { conversation }
18  </dialogue>
19
20  DO NOT ask any question that has been asked in the  dialogue above.
21
22  Your question  does  not have to be entirely  decided by the  belief . You can construct any question that make yourself more confident about what the
        image is .
23  Think step by step and reason about your  uncertainty  of the image to generate . Make sure to ask only one question . Make sure it  is  not very
        difficult  for the user to answer. For example, do not ask a very very long question , which can take the user a long time to read and answer
        .
24  Make sure that  you question the answer within <question> and </question> markers.
```

## D.15  HSA QUESTION PROMPT INSTRUCTION

```
1   You are  constructing  a text –to–image (T2I) prompt and want more  details  from the user .
2   You have to ask a question about the the most important  entity  or the  attribute  of the most important  entity .
3   We have entity  types : ( i ) explicit : directly ask question with options ; ( ii ) implicit : ask whether this  entity  required for the image with yes or
        no as options ; ( iii ) background: ignore the  attribute  value and directly ask the value of the entity . ( iv ) relation : add keyword like '
        relation ' to emphasize this  entity  is a relation .
4   Construct a simple question that  directly  asks this information from the user and also  provides  option that the user can pick from. Ask only one
        question and follow  it  with  options .
5
6
7   Example1:
8    entity : rabbit
9    attribute : color
10   candidates : black, white, brown
11   entity_type : explicit
12   question : What color of the  rabbit  do you have in mind? a. black, b. white, c. brown. d. unkown. If none of these  options , what color of the  rabbit
        do you have in mind?
13
14   ...  [[ a few additional  examples]]  ...
15
16   Example5:
17   entity : $entity
18   attribute : $attribute
19   candidates : $candidates
20   entity_type : $entity_type
21   question :
```

---

**Algorithm 2** User-Agent Self-Play Algorithm

---

1: **Input:** Initial prompt $p_0$, User $u$, Agent $a$ (with $p_0$), $max\_turns$
2: **Output:** Refined prompt $p_f$
3: $p_f \leftarrow p_0$
4: **for** $turn\_id = 0$ **to** $max\_turns - 1$ **do**
5:     $action \leftarrow a.\text{SelectAction}()$
6:     $question \leftarrow a.\text{VerbalizeAction}(action)$
7:     $answer \leftarrow u.\text{AnswerQuestion}(question)$
8:     $a.\text{Transition}(action, answer)$
9:     $p_f \leftarrow a.prompt$
10: **end for**
11: **return** $p_f$

---

## E  BACKGROUND

**T2I Evaluation.** Inception Score (Salimans et al., 2016) and Frechet Inception Distance (Heusel et al., 2018) are popular metrics to measure the fidelity of generated images, i.e. the similarity of the generated images to real ones. Improved precision and recall (Kynkäänniemi et al., 2019) allows to analyse the sample quality and the coverage independently. Since text prompts are used to guide image generation in T2I models, image-prompt alignment is an important evaluation metric which can be classified as embedding-based such as CLIPScore (Hessel et al., 2022), ALIGNScore (Zha et al., 2023), VQA-based metrics such as TIFA (Hu et al., 2023), DSG (Cho et al., 2023) abd VQAScore (Lin et al., 2024) and captioning based metrics like LLMScore (Lu et al., 2023). Approaches such as PickScore (Kirstain et al., 2023), ImageReward (Xu et al., 2023) and HPS-v2 (Wu et al., 2023) finetune models on human ratings to devise a metric that aligns with human preferences. Recently, diversity of generated images (Naeem et al., 2020) is becoming an important metric of measurement to track progress, especially in the geo-cultural context (Kannen et al., 2024; Hall et al., 2024). Diversity can be used to quantify the under-specification in the input prompt: more specific the prompt, the less diverse are the generated set of images across different seeds (Kannen et al., 2024).

**Prompt expansion** is a widely known technique to improve image generation (Betker et al., 2023). ImageinWords (Garg et al., 2024) proposes to obtain high-quality hyper-detailed captions for images, which significantly improve quality of image generation. Datta et al. (2024) present a generic prompt expansion framework used along Text-to-Image generation and show an increase in user satisfaction through human study. Our work can be viewed as a method to adaptively expand a T2I prompt based on user feedback. Samples from the agent belief can be used to construct expanded prompts.

## F  AUTOMATED EVALUATION

In Algorithm 2, we show the user-agent self-play procedures that we used to perform all automated evaluation.

## G  DETAILS ON THE AGENT INTERFACE

Below is a showcase of how users could interact with the belief graph and clarifications in a hypothesised interface, to better iterate their inputs, to reach higher a quality and satisfaction of outputs. This is a crudely hypothesised, intentionally simple interface for the sake of research, but could be iterated and improved upon in many ways depending on application and users.

**1. Default state** On load of the app, there would be a text prompt input and space for output images, as is common across typical T2I interfaces. There would also be space for the user to view either clarifications from the model, or a graph interface, as part of the overall "input" section as these would act as a further input for future model output iterations. See Figure 11 below as reference.

**2. Output images, with Clarifications** Once the user has submitted the prompt and the model has responded, there would be a set of images, as initial outputs from the users prompt. Below the input prompt would be a set of "Clarifications" in its populated state. These clarifications would ask the

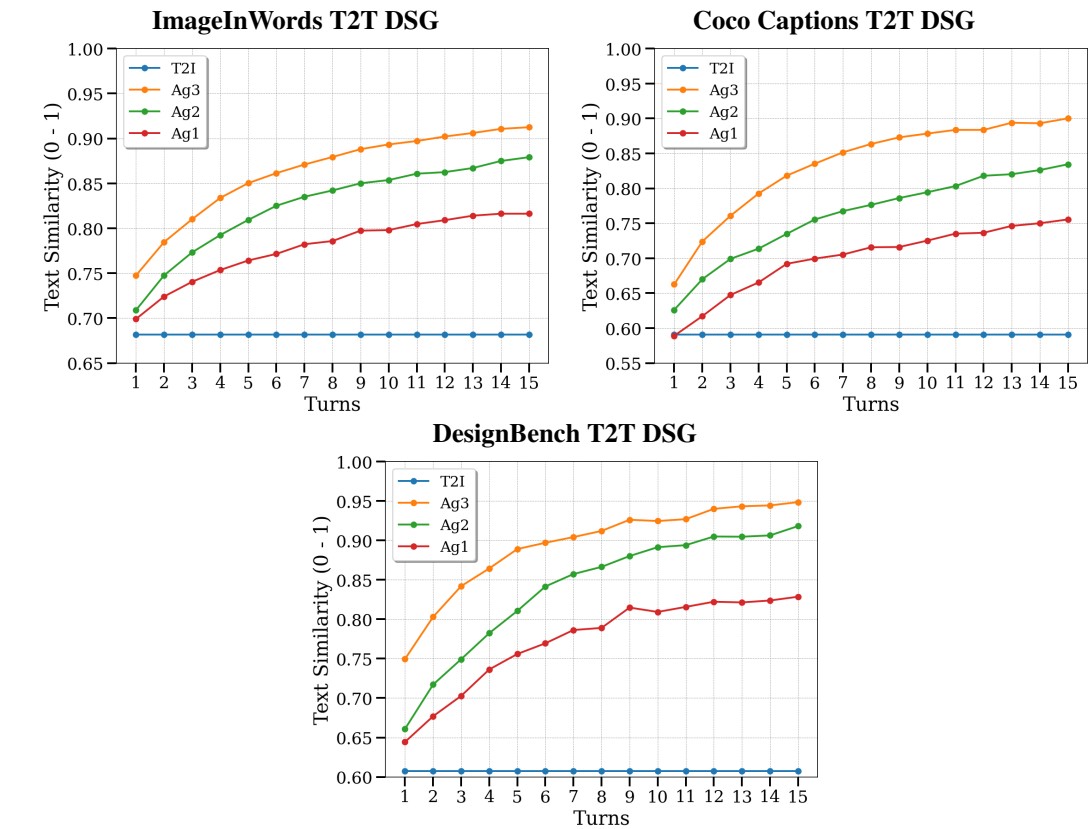

Figure 10: DSG score comparison between ground truth prompt and agent generated prompt reported at each turn. The performance of all agents increase with increase in number of turns.

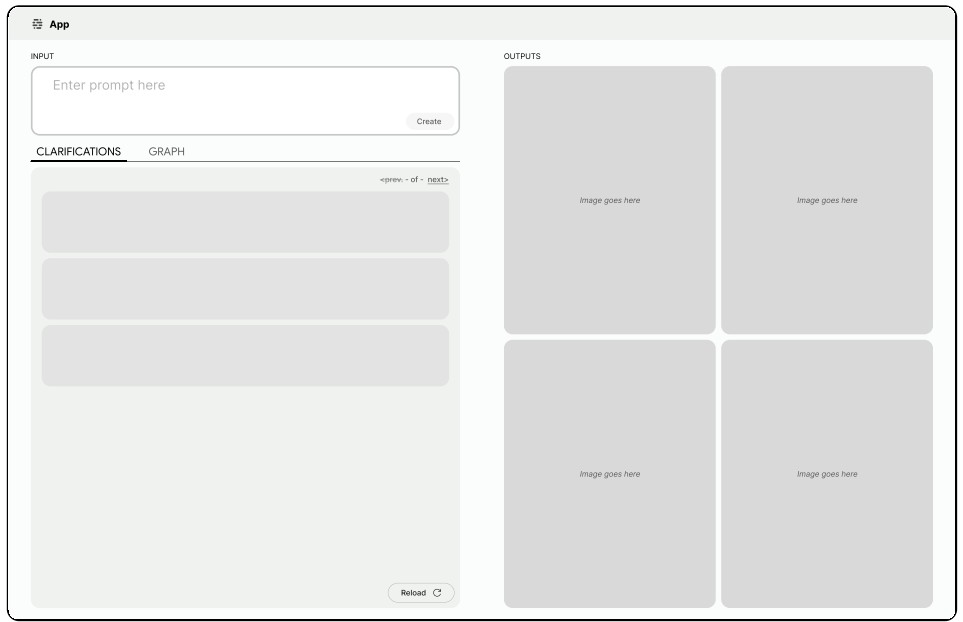

Figure 11: Default state of a possible interface.

user specific questions that would be necessary to increase the specificity of the prompt, for the model to get a more accurate results aligned to the users intention, or to help the user realise their intention.

Options would be given of the highest probability options for each Clarification, but the user could also fill in a totally new option via a free text field. Once answered by selection or text input, the clarifications would be added to the above, primary prompt for regeneration when the user selects. See Figure 12 below as reference.

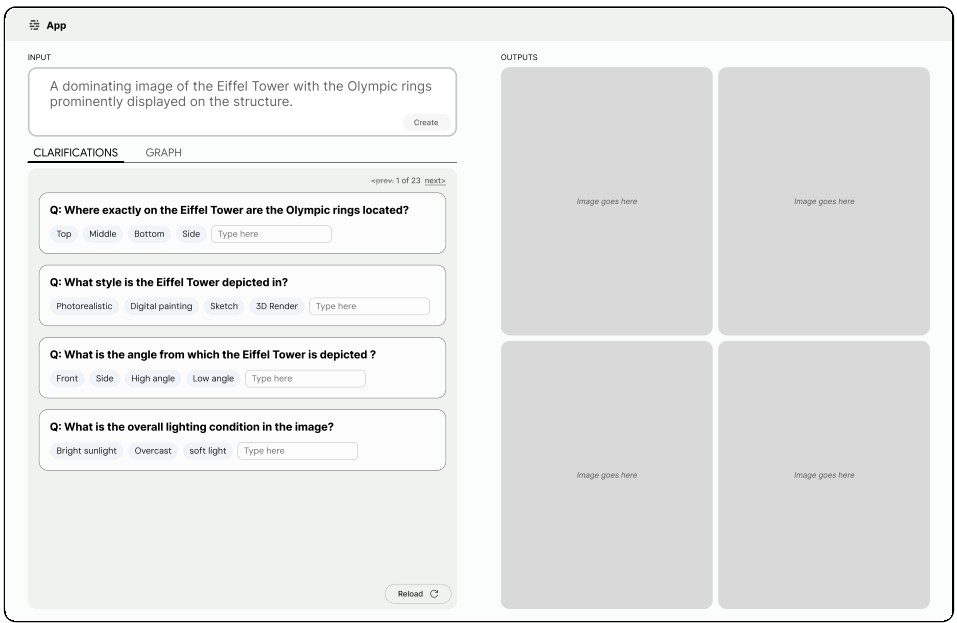

Figure 12: Interface once prompt has been input with clarifications.

**3. Graph Entities & Attributes** Instead of the clarifications, the user could select to instead view a Graph by clicking Tab above the clarifications themselves. This graph would be populated will all Entities from the prompt explicit and implicit visually defined differently (in this diagram by the dotted line surrounds implicit entities, but is a filled line when surrounding explicit entities). The graph layout will be structured, depicting relationships concentrically i.e. "on", "in" or "under" for example, will become child entities, and be displayed within the parent entities' boundary. For example a 'Mug' that has the relationship of 'on' a 'Table' entity, will sit within the boundary of 'Table', as also would a 'Plate' if that had the same child-parent relationship.

Below the Graph would also be a list of 'cards' (i.e. boxed groups of information), one for each "explicit" or "implicit" entity. Within each card a user could see the status of implicit / explicit, and change this status to confirm or deny its presence. The user could also see a list of "attributes" associated to that entity, which the model has assumed. Each of these attributes could be changed by interacting with a list of alternatives via drop down. These lists are determined in terms of which items and order of items, based on the probability by which the model sees them, ordered with higest first. This probability would be made clear to the user to define the order by seeing the peercentage next to the label. See Figure 13 below as reference.

**4. Graph Relationships** The user would also be able to change the state of the Graph and Cards, to instead focus on the relationships between entities, by toggling to "Relations". In this state the user would be able to focus on two specific entities (e.g. 'mug' and 'table'), see the description of the relationship (e.g. 'the mug is sitting on the table') and if desired change the relationship to an alternative (e.g. 'on', changed to 'under') via a drop down of options which the model determined as alternative options ordered by probability, as per attributes. See Figure Figure 14 below as reference.

Once any of these changes are made the user could initiate a regeneration via the updated prompt to create a new set of output images, which can then be further refined via the same method.

## H  DETAILS ON USER STUDY

Below we describe the exact guideline definitions we shared with the user for a user study.

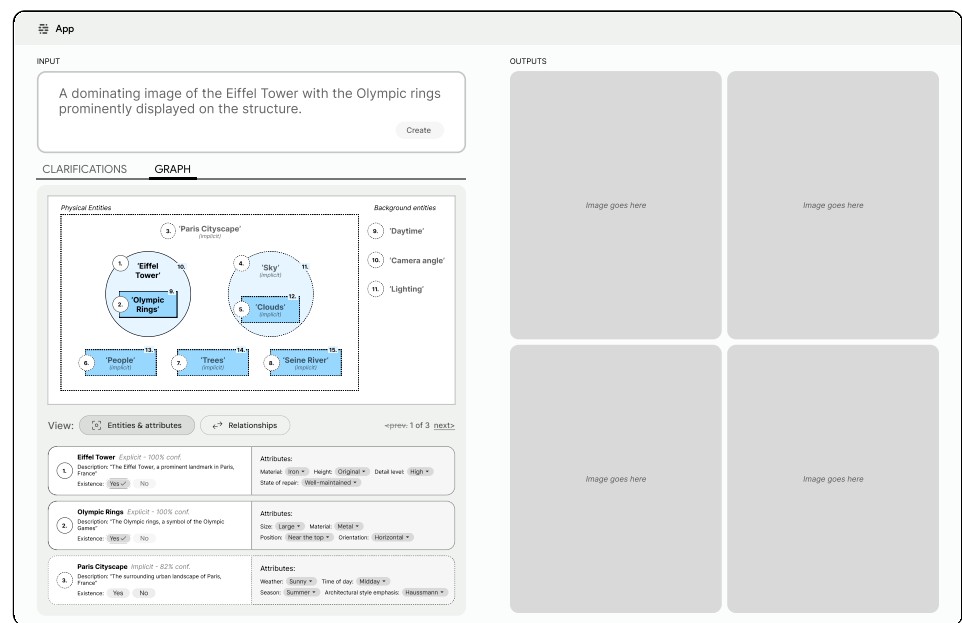

Figure 13: Interface with Graph displaying Entities, with cards below enabling a user to change attributes associated to each entity.

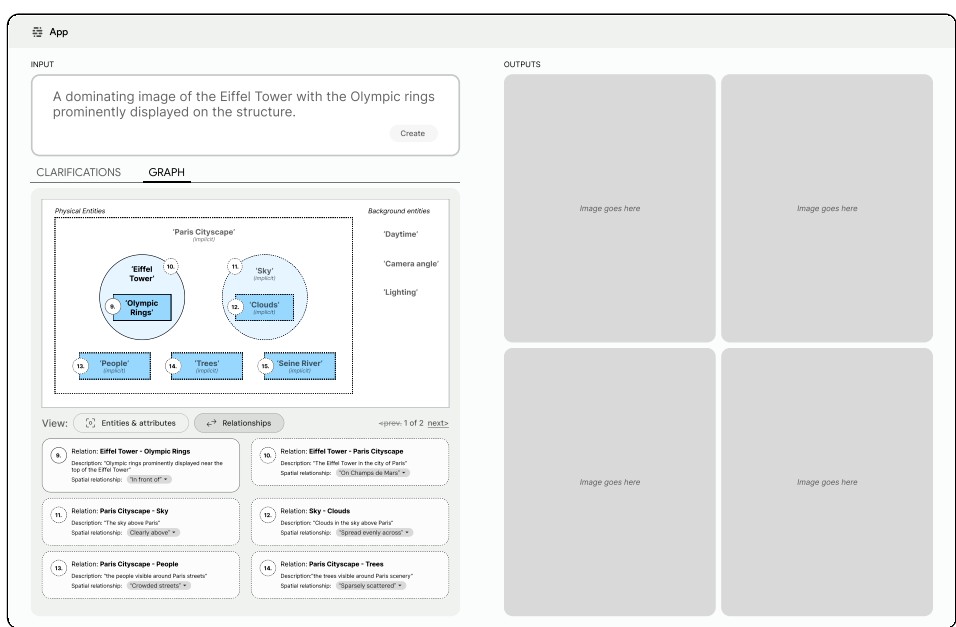

Figure 14: Interface with Graph displaying relations between Entities, with cards below enabling a user to change relationships between entities.

### H.0.1 HYPOTHESIZED FRUSTRATIONS

We presented participants with the following hypothesized frustrations related to T2I model usage:

1. **Prompt Misinterpretation:** The model misunderstands complex relationships between entities in the input prompt.

2. **Many Prompt Iterations:** The model does not immediately generate what the user intends, requiring numerous iterative changes to the input prompt.

3. **Inconsistent Generations:** The model reinterprets the input prompt differently between iterations, causing unwanted changes in the generated images.

4. **Incorrect Assumptions:** The model makes incorrect assumptions or no assumptions when encountering gaps in the details provided in the input prompt, leading to undesired outputs.

Explanations of terms were given to users of:

1. "Entities" are single items that are intended to be in the image e.g. "Cat" and "Ball", from "make a sketch of a Cat playing with a Ball"

2. "Prompt" means the text written to communicate the intended output image e.g. the sentence "make a sketch of a Cat playing with a Ball" is the "Prompt", also known as "Input"

3. "Iterations" are each set of different image outputs by the model, taken from a different input, or even the same input just regenerated

The question asked for each Frustration were: "Please score the below frustrations (or issues) that could be related to Text to Image AI Generation"."Rank in terms of how much they relate to your current usage, with your most commonly used model or app."

### H.0.2    HYPOTHESIZED FEATURES

We proposed the following features as potential solutions to address the identified frustrations:

1. **Clarifications:** The model would ask specific clarifying questions about uncertainties in the prompt. These details would then be incorporated into subsequent iterations. For example: "Is the cat playing with: 1. a ball of wool, or 2. a tennis ball?"

2. **Graph of Prompt Entities:** A visual representation of all entities in the prompt as a graph, allowing users to see and edit attributes of each entity. E.g., seeing that the model has assigned "round," "small," and "wooden" as attributes to "table" and allowing the user to change them to "square" and "metal."

3. **Graph of Prompt Relationships:** A visual representation of relationships between entities in the prompt, allowing users to see and edit these relationships. E.g., seeing that "donut" is "next to" "coffee" and allowing the user to change the relationship to "on top of."

The questions asked for each feature were:

1. "How likely this feature is to help your current workflow if you had it now?". With response options of: "Very unlikely to help", "Unlikely to help", "Could help", "Likely to help", "Very likely to help".

2. "How soon would this feature deliver value to your work?" with response options of: "Very soon / immediately", "Sometime, "Not very soon".

Image references were given for each Feature as listed out below:

1. **Clarifications:**

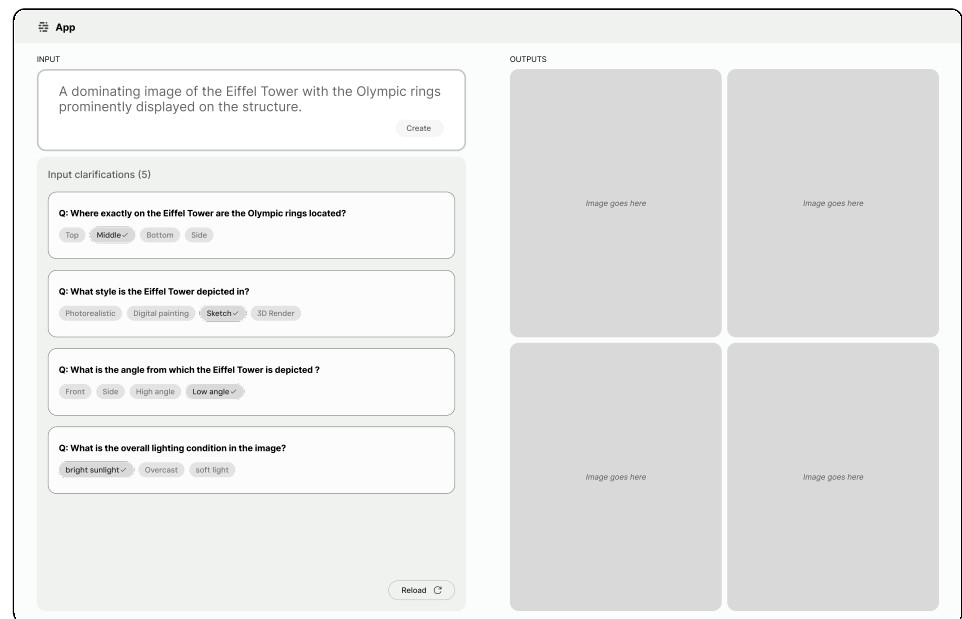

Figure 15: Stimulus image in the survey to test the Model clarifications feature.

2. **Graph of Prompt Entities:**

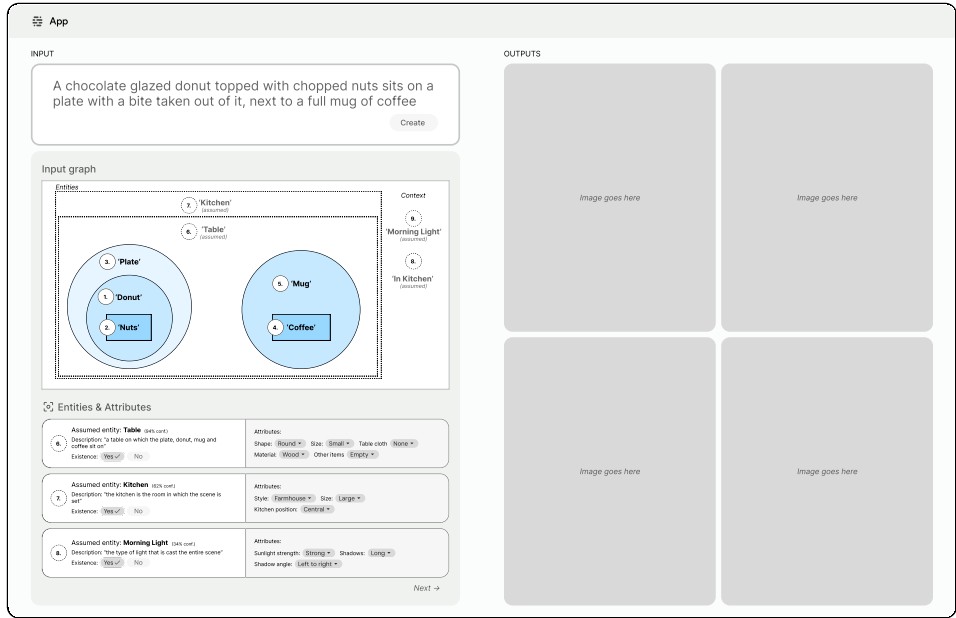

Figure 16: Stimulus image in the survey to test the Model Graph of Entities and Attributes feature.

3. **Graph of Prompt Relationships:**

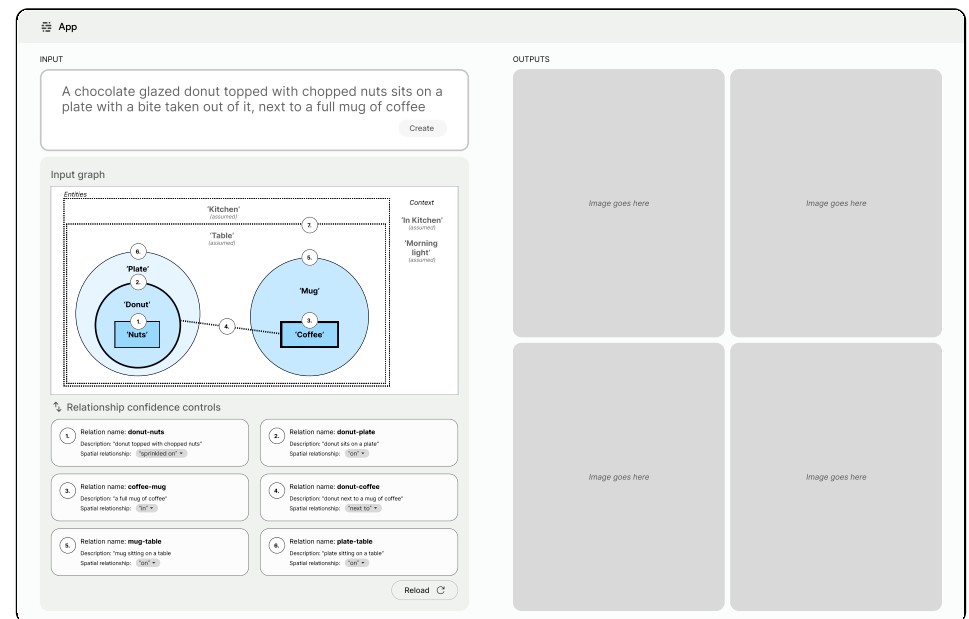

Figure 17: Stimulus image in the survey to test the Model Graph of Entity Relations feature.

### H.0.3 HUMAN STUDY RESULTS

Table 3: Breakdown of the T2I usage frequency of the 143 participants recorded

| Usage Frequency | No. of users | (%) |
|---|---|---|
| Many times a day | 13 | 9.1 |
| Many times a week | 44 | 30.8 |
| At least once a week | 36 | 25.2 |
| At least once a month | 50 | 35.0 |

Table 4: Reported User Frustrations with existing T2I processes (% of users)

| Frustration | V. Freq. (%) | Freq. (%) | Occas. (%) | V. Occas. (%) | No Issue (%) |
|---|---|---|---|---|---|
| Prompt Misinterpret. | 7 | 19.6 | 43.4 | 23.1 | 7 |
| Many Iterations | 10.5 | 44.8 | 28 | 11.9 | 4.9 |
| Inconsistent Gen. | 11.2 | 20.3 | 39.9 | 21 | 7.7 |
| Incorrect Assumptions | 7 | 23.1 | 39.2 | 20.3 | 10.5 |

Table 5: Expected speed of value delivered from features (% of users)

| Feature | Very soon / immediately (%) | Sometime(%) | Not very soon. (%) |
|---|---|---|---|
| Clarifications | 57.7 | 37.2 | 5.1 |
| Entity Graph | 49.6 | 34.8 | 15.6 |
| Relation Graph | 41.8 | 44 | 14.2 |

