# OpenReview forum: "Proactive Agents for Multi-Turn Text-to-Image Generation Under Uncertainty"
_ICLR.cc/2025/Conference — Submitted to ICLR 2025_

### Official Review · Reviewer_Tosu · 2024-11-02

**Soundness:** 3
**Presentation:** 2
**Contribution:** 2
**Rating:** 5
**Confidence:** 4

**Summary:**

This paper introduces a proactive design for text-to-image (T2I) agents that enhances user interaction by prompting clarification questions when user intent is unclear. The approach enables T2I agents to create an interpretable 'belief state'—a representation of their current understanding—that users can review and adjust as necessary. By allowing edits to this belief state, the system promotes transparency and collaboration, enabling the agent to refine its responses to better match the user’s expectations. Overall, this design aims to make T2I agents more interactive, adaptable, and user-centric.

**Strengths:**

The paper is well-written and easy to understand, presenting concepts and methodologies in a clear and accessible manner. Additionally, the results are strong, demonstrating the effectiveness and reliability of the proposed approach.

**Weaknesses:**

However, many previous methods have proposed self-correction techniques to improve generated images, as demonstrated in works such as [1], [2], and [3]. This paper, however, does not include comparisons with these related methods. Given this omission, I question the technical contribution of the paper and am uncertain whether it would be a suitable fit for CVPR.

Previous related works have proposed iterative improvements to image quality. Therefore, it would be helpful if the authors could address the following points:
1) What is the difference between the iterative procedure used in the proposed methods and that of related works? There is no direct comparison with all related works.

2) It would be beneficial if the authors could compare their methods with these works and explain why the proposed method has a competitive edge.

3) If users make errors when inputting their data, is the proposed model able to detect and correct these mistakes?

4) Regarding reasoning ability, if a user requests advanced generation requirements, such as "a robot executes a task," is the proposed method capable of handling spatially-aware generation?

[1] Wu, Tsung-Han, et al. "Self-correcting llm-controlled diffusion models." Proceedings of the IEEE/CVF Conference on Computer Vision and Pattern Recognition. 2024.

[2] Yang, Ling, et al. "Mastering text-to-image diffusion: Recaptioning, planning, and generating with multimodal llms." Forty-first International Conference on Machine Learning. 2024.

[3] Jiang, Dongzhi, et al. "CoMat: Aligning Text-to-Image Diffusion Model with Image-to-Text Concept Matching." arXiv preprint arXiv:2404.03653 (2024).

**Questions:**

It would be appreciated if the authors could elaborate on the technical challenges addressed, as well as the potential insights and future impact of this work in the field of computer vision.

=========================

After reviewing the rebuttal, I have decided to keep my score unchanged.

---

> ### Author Response · Authors · 2024-11-20
>
> >  uncertain whether it would be a suitable fit for CVPR
>
> This is a ICLR 2025 submission and not a CVPR submission. Our main contributions are not primarily within the computer vision space but in AI/ML. Could you please recalibrate the scores with this information?
>
> >  I question the technical contribution of the paper
>
>
> Please see the general reply for a summary of the technical contributions.
>
>
> > many previous methods have proposed self-correction techniques to improve generated images, as demonstrated in works such as [1], [2], and [3]...
>
> The papers [1, 2, 3] focus on improving the alignment of T2I models to input prompts. **Whereas this paper focuses on iterative clarification/refinement of an under specified prompt.** These are two very different objectives and while both very important, this paper focuses on the latter objective.
>
> > What is the difference between the iterative procedure used in the proposed methods and that of related works? There is no direct comparison with all related works.
>
> As mentioned above, the related works the reviewer listed and our work solve very different problems. Those related works aim to develop better methods to perform the T2I task itself, assuming the text description is detailed enough to be used to evaluate the quality of the generated images, so that they can iteratively generate better images based on the same text description.  While in our work, we tackle that challenge of how to actively gather information from the user in an interactive way, when the user doesn’t write prompts that fully specify the image they would like to generate. Given that the problems are so different, the solutions are not comparable. While both methods are iterative, they are iterative in entirely different ways: we have the user in the loop. Hence, we believe it is not appropriate to compare our work with the cited papers. We assure to discuss this distinction in the related work.
>
> > It would be beneficial if the authors could compare their methods with these works and explain why the proposed method has a competitive edge.
>
> We will add the clarification stated above to the paper along with a reference to the mentioned works, however we are unable to compare them as they do not study the same problem as we do in this paper.
>
> > If users make errors when inputting their data, is the proposed model able to detect and correct these mistakes?
>
> We assume the reviewer is asking if the user can update the answers to previous questions in the case that the user changes their mind - ( Ie. they input “the house is red” but then decide they want to change the house to blue color) but kindly let us know if we misunderstood. The answer is yes the user can choose to directly edit the prompt, directly edit the belief graph or simply input a sentence to the agent describing their updated decision on the attributes of the image.
>
> > Regarding reasoning ability, if a user requests advanced generation requirements, such as "a robot executes a task," is the proposed method capable of handling spatially-aware generation?
>
> We are not entirely sure what the reviewer meant by “reasoning ability” and “advanced generation requirements” in this context. We assume the reviewer is asking whether our T2I agent is able to identify and incorporate spatial relations mentioned in a prompt, a clarification question/answer or the belief graph. The answer is yes. In fact, a critical part of the belief graph is the spatial relations between entities in our design of the agent.

---

> > ### Author Response · Authors · 2024-11-20
> >
> > > It would be appreciated if the authors could elaborate on the technical challenges addressed, as well as the potential insights and future impact of this work in the field of computer vision.
> >
> > The technical challenges we addressed lie in the difficulty of enabling effective and efficient human-AI interaction. The challenges include 1) system design for proactive T2I agents, 2) interpretable, transparent, and controllable belief states, 3) automated evaluation of proactive T2I agents. Please see the general reply for the detailed novelty and contributions.
> > While this work does not specifically aim to improve computer vision models, the potential insights and future impact to the CV field include:
> > 1. Incorporating agency in computer vision systems can enable better interaction with humans and optimize user experience.
> > 2. Belief states and clarification questions are useful for better T2I experience for human users.
> > 3. We believe that improving model quality alone (e.g., text-image alignment from a computer vision perspective) is not enough for deploying useful AI tools. We also need to consider how to build methods and interfaces that enable a quick exchange of information between humans and AI, so that the system can efficiently generate the content the user is actually looking for without too much back and forth. The visual information exchange problem between human and CV models can potentially be an important future direction in the field of CV as more researchers are interested in building human-AI collaboration systems.

---

> > > ### Author Response · Authors · 2024-11-26
> > > **Clarifying reasons for keeping the rating**
> > >
> > > Dear reviewer Tosu,
> > >
> > > Could you please elaborate on the reasons for keeping the score unchanged? Please note **that this conference is not CVPR**. We would really appreciate a recalibration of the rating given our strong technical contribution to the human-AI interaction subfield of AI/ML.
> > >
> > > Thank you in advance for your understanding.
> > >
> > > -Authors

---

### Official Review · Reviewer_JLtN · 2024-11-03

**Soundness:** 3
**Presentation:** 3
**Contribution:** 3
**Rating:** 6
**Confidence:** 4

**Summary:**

This paper introduces a proactive approach to multi-turn text-to-image (T2I) generation, addressing the common problem of prompt underspecification, where users struggle to articulate their vision clearly, leading to suboptimal results. The proposed T2I agents actively engage with users by asking clarification questions and updating their understanding through an interpretable belief state that users can edit. The paper details the design of these agents, the use of a graph-based belief state to manage uncertainty, and the development of a new benchmark, DesignBench, for evaluation. Experimental results, including both automated evaluations and human studies, demonstrate that these proactive agents significantly improve image generation quality, achieving higher alignment with user intent compared to traditional single-turn T2I models.

**Strengths:**

The paper's strength lies in its interactive approach to improving text-to-image (T2I) generation. By using proactive agents that ask clarification questions and utilize a graph-based belief state for transparency, the method addresses prompt underspecification effectively. The combination of user-centric design, comprehensive evaluation, and the introduction of the DesignBench benchmark demonstrates significant improvements over traditional T2I models.

**Weaknesses:**

1. If the approach involves making prompt engineering more detailed than the initial prompt, it seems intuitive that the resulting image would be more accurately generated as intended. Could you elaborate on what aspects of this method are novel or distinct beyond providing more detailed prompts?
2. How do you determine which characteristics are crucial for accurately generating each image? A clearer and more logical explanation of the criteria or process used for this selection would be helpful.

**Questions:**

1. Is there a validation process for the ground truth caption? What guarantees that it’s a well-generated caption?
2. I wonder if 15 turns are essential. The number of turns needed would likely vary for each case, so wasn’t this considered? Why 15 turns specifically?

---

> ### Author Response · Authors · 2024-11-20
>
> > If the approach involves making prompt engineering more detailed than the initial prompt, it seems intuitive that the resulting image would be more accurately generated as intended. Could you elaborate on what aspects of this method are novel or distinct beyond providing more detailed prompts?
>
> Current prompt expansion techniques typically aim to make images more aesthetically pleasing or diverse [1]. Our work can be considered as an approach for agents to actively gather information from users so that the prompt can be adaptively and sequentially expanded in a way that aligns with user intents. The followings are some aspects that we consider but current prompt expansion techniques do not:
> - how to reduce the uncertainty a model has about underspecified user prompts;
> - generating clarification questions / belief graphs and having users in the loop to answer clarification questions or manipulate model beliefs;
> - adaptively refining the prompt based on interactions with the user to align with user intents.
>
> Please see the general reply for the full description of novelty and contribution.
>
> [1] Siddhartha Datta, Alexander Ku, Deepak Ramachandran, Peter Anderson. Prompt Expansion for Adaptive Text-to-Image Generation. 2023. https://arxiv.org/abs/2312.16720
>
> > How do you determine which characteristics are crucial for accurately generating each image? A clearer and more logical explanation of the criteria or process used for this selection would be helpful.
>
> We are not entirely sure what the reviewer is asking and have two hypotheses on what the reviewer means: (1) “characteristics” refers to characteristics of the image, and the reviewer is asking which characteristics are crucial for determining if the image is accurate and why; (2) “characteristics” refers to characteristics of the thing to be clarified, and the reviewer is asking about the criteria for determining what is should to be clarified (in order to generate accurate images) and why. Please let us know if we misunderstood.
>
> (1) which characteristics are crucial for determining if the image is accurate and why:
>
> The important characteristics are 1) the inclusion of the entities and their attributes that the user intends to generate, 2) the image reflects the relations between entities that the user intends to generate, 3) the image is semantically consistent with the ground truth description of the image (the ground truth description describes in detail what the user intends to generate).
>
> Logical explanation on why we believe these characteristics matters:
>
> The crucial characteristics for accurate image generation are reflected by T2I metrics. The two well-known metrics are DSG scores and VQA scores. DSG focuses on composition (e.g., entities, attributes) and spatial relations of the generated image, whereas VQA focuses more on overall semantic understanding of the generated image in relation to the prompt. If the above characteristics are met, both scores will be high, meaning that the image accurately reflects user intents.
>
> (2) the criteria for determining what should be clarified and why:
>
> We detailed the criteria of what clarification questions should be asked in Section 4.1. We constructed 3 agent prototypes and each has its own question-asking criteria, which the reviewer can find in Section 4.1.2 and more details in the corresponding appendix. For example, the characteristics Ag1 uses in its question-asking criteria include (a) the uncertainty the agent has about an entity’s existence, the value of an attribute for an entity, and relation between entities, (b) the importance of the entity, attribute or relation relevant to the prompt. Both characteristics are part of the belief graph, which is generated by LLMs using in context learning.
>
> Why use those characteristics? The belief graph directly captures the composition and spatial relations considered in DSG, since it includes possible entities, their attributes and spatial relations between entities. (a) If the agent is very uncertain about the existence of an entity, it is likely that the DSG score can increase once the user provides the information of whether the entity exists. Similarly for attributes and relations. (b) If the agent is uncertain about an attribute of an entity but the attribute is not so important, the agent probably shouldn’t ask about that entity, since the ground truth description probably does not even mention that attribute. For example, for prompt “a red hat”, the agent can be unsure about the exact shade of red, but the exact shade likely does not matter since the user might like any red color.
>
> To make sure these characteristics are indeed crucial, we verified the performance of those agents in experiments. All agents achieve higher VQA and DSG scores than the standard T2I approach.

---

> > ### Author Response · Authors · 2024-11-20
> >
> > > Is there a validation process for the ground truth caption? What guarantees that it’s a well-generated caption?
> >
> > This is an important question and to add quantitative validity to the ground truth caption generation we perform Text to Image (VQA) Similarity between the ground truth caption and the ground truth over all images in the DesignBench dataset and report the mean, median and std. DesignBench GT Caption to GT Image T2I Vqa similarity:  mean 0.9999998529754514, median 1.0, std 4.541245460313302e-07. The mean is extremely close to 1 as we would expect a good caption to be and we can compare this number with the T2I results in Table 2 for the T2I model and 3 agents.
> >
> > We will add this result to the paper as well as the same experiment for all datasets.
> >
> > > I wonder if 15 turns are essential. The number of turns needed would likely vary for each case, so wasn’t this considered? Why 15 turns specifically?
> >
> > We evaluated different numbers of turns in early experiments with text-to-text similarity and found that around 8 turns we see some plateau improvement similarity scores (this can be seen in Figure 3, Graph a). To accommodate the variability of difficulty among images, we extended the number of turns up to 15. In the human user studies we conducted (L504), we found that most 60% of people disliked the frequency of prompt refinement in traditional T2I UIs. Therefore a goal in our proposed agentic method was to achieve high fidelity in a short number of turns. We believe that 15 turns allows for the variance of difficulty between images while constraining the agent’s maximum number of turns - thus forcing the line of questioning to be efficient. Moreover, our evaluation approach is not restricted to 15 turns, so if anyone adopts our evaluation approach in the future, they can use any number of turns.

---

> > > ### Comment · Reviewer_JLtN · 2024-11-26
> > >
> > > Thank you for your response. I decided to keep my score unchanged.

---

> > > > ### Author Response · Authors · 2024-11-26
> > > >
> > > > Dear Reviewer JLtN,
> > > >
> > > > Thank you for your confirmation. Could you please let us know if you have any remaining concerns that we did not address? What changes would you like us to make to make our work stronger?
> > > >
> > > > We are also wondering if our interpretations of your questions are correct.
> > > >
> > > > -Authors

---

> > > > > ### Comment · Reviewer_JLtN · 2024-11-27
> > > > >
> > > > > 1. Many studies [1] focus on directly editing images rather than relying on prompt engineering to generate images that align with the user's intent. These approaches are likely gaining attention because they are more intuitive and effective compared to prompt engineering. Is there no comparison with these methods? If the goal of this paper is indeed to generate images that align with user perspectives, such a comparison seems essential.
> > > > >
> > > > > [1] Self-correcting LLM-controlled Diffusion Models, CVPR '24
> > > > >
> > > > > 2. Another question that arises is, why not input a detailed prompt from the beginning? If a prompt contains sufficient information about the goal image, wouldn't it be possible to generate a good image from the start? If multi-turn interaction isn't strictly necessary because sufficiently detailed prompts still result in images different from the goal image, the proposed method lacks convincing justification for its necessity.
> > > > >
> > > > > 3. Furthermore, in many cases, no matter how well the prompt is crafted, the generated image may not turn out as intended. Doesn't this method fail to address such situations?
> > > > >
> > > > > 4. Lastly, the criteria for determining what is uncertain seem ambiguous. Why is the color of a rabbit important during generation? If the goal is to generate objects suited to each country or cultural context, adjustments would be needed for many properties, such as ears, eyes, nose, and tail, in addition to color. This paper does not appear to exhibit such editing capabilities.

---

> ### Author Response · Authors · 2024-12-01
>
> > 1. Many studies [1] focus on directly editing images...
>
> The paper [1] and our work solve very different problems. [1] focuses on improving the alignment between generated images and input prompts via iterative self-correction. [1] assumes that the input prompt is detailed enough such that it is possible to assess the alignment and perform self correction. [1] mentions image editing as a side product, but it relies heavily on the user to know the “language” for editing (e.g., Move up “monkey #1”) which is used for self-correction. Our work, on the other hand, aims to design the overall human-AI interaction framework with T2I as a case study (see 1st paragraph of introduction). Our work focuses on **proactive information gathering to clarify an underspecified prompt and belief graph for interpretability**, which we believe to be generally useful for agentic behaviors during human-AI interaction.
>
> There exists other image editing works such as Instruct-Imagen, which passively accepts new instructions from users, as opposed to actively collecting information from users. Such editing methods can be included in our human-AI interface to provide a diverse set of controls for users. But in essence, image editing and our work are two entirely orthogonal modalities of the T2I system. Image editing fundamentally aims to improve the model (such that it can accept new instructions) and we aim to provide agency for T2I. Hence we don’t think our work and image editing are comparable.
>
>
> > 2. ...why not input a detailed prompt from the beginning? If a prompt contains sufficient information about the goal image, wouldn't it be possible to generate a good image from the start? If multi-turn interaction isn't strictly necessary because sufficiently detailed prompts still result in images different from the goal image, the proposed method lacks convincing justification for its necessity.
>
> We agree that sometimes multi-turn interaction is not strictly necessary if the prompt has sufficient details. However, in our user study, we observed that 54.5% of human subjects found many prompt iterations to be a frequent issue (“the model does not immediately generate what I want, so I have to iteratively change my input prompt many times”). Only 5.4% found no issues with prompt iterations. Moreover, at least 90% of human subjects found our agents and their belief graphs helpful for their T2I workflow. (see both abstract, intro and section 5.4) This means our agentic framework will be useful for the majority of potential T2I users. We believe this is a strong justification for why our work is necessary.
>
>
> > 3. Furthermore, in many cases, no matter how well the prompt is crafted, the generated image may not turn out as intended. Doesn't this method fail to address such situations?
>
> Yes, our proposed framework does not address the prompt-image alignment problem, but studies the problem of iterative clarification/refinement of an underspecified prompt. Those two are two very important and orthogonal objectives.
>
> The agent prototypes we described in the paper are highly modular: the agents use frozen T2I models to generate images based on the prompts that the agent updated. Therefore the quality of the final images are directly influenced by the ability of the T2I model employed. This also means that when a better off-the-shelf T2I model becomes available, it can be directly plugged into the agents and the system will achieve better performance without any additional adaptation.
>
> Even with a highly detailed and unambiguous prompt, the system may not produce a perfect image if the underlying T2I model is inadequate. Conversely, even with a perfect T2I model, uncertainties or ambiguities in the prompt can hinder the system's performance. Many works aim to address the issue of T2I prompt to image alignment [1,2,3,4,5,6] but almost none for the second in the literature of T2I. Our work focuses on the prompt underspecification problem and aims to design agents that proactively ask users clarification questions and present beliefs to gather information about what the user wants to generate. The belief graph and the final prompt of our proposed agent could have accurate entities, attributes and relationships and in some cases T2I agents fail to follow parts of the prompts or create distorted assets in the images [4].
>
> Table 1 and Table 2 we show evaluation metrics over T2T, T2I and I2I similarity. We can see the T2T scores as a form of system ablation, since we exclude the T2I model and only perform similarity on the caption input to the T2I model. This removes any error or prior information that the T2I model adds. In Table 1 and Table 2 per the T2T metrics, we have achieved a 92%+ similarity score, and therefore a better T2I model for the agent can immediately boost the performance of the overall agent system.

---

> > ### Author Response · Authors · 2024-12-01
> >
> > > 4. Lastly, the criteria for determining what is uncertain seem ambiguous. Why is the color of a rabbit important during generation? If the goal is to generate objects suited to each country or cultural context, adjustments would be needed for many properties, such as ears, eyes, nose, and tail, in addition to color. This paper does not appear to exhibit such editing capabilities.
> >
> > We believe the reason the agent decides to ask about color is that knowing the color can reduce the uncertainty about other aspects of the rabbit. (The importance scores are automatically generated by LLMs using ICL and instructions that stress principles such as Relevance, Uncertainty Reduction and Easy-to-Answer in Section 4.1). For example, if the rabbit is reddish-brown or grayish-brown, the rabbit is most likely to be a cottontail rabbit, and all other aspects of the rabbit including ear, eye, nose etc would be immediately determined. In other words, asking about color or breed (which often appear as well) first can improve the communication efficiency with users, which is the objective that an agent is trying to achieve and what we evaluated in the experiments.
> >
> > To suit each country and cultural context, the agent should ask the question that can reduce uncertainty about the cultural aspect of the user the most. The alignment to user intent is designed to be sequential and iterative based on agent beliefs. If the agent has an initial belief about the possible user backgrounds, it can potentially ask other questions that reduce the uncertainty the most about other aspects of the user that impacts their desired image. Autonomously building in the initial belief (i.e., the prior) of the agent is a very important and very challenging problem in Bayesian machine learning [7]. A full investigation on this topic is not going to fit a conference paper.
> >
> >
> > [1] Wu, Tsung-Han, et al. "Self-correcting llm-controlled diffusion models." Proceedings of the IEEE/CVF Conference on Computer Vision and Pattern Recognition. 2024.
> >
> > [2] Yang, Ling, et al. "Mastering text-to-image diffusion: Recaptioning, planning, and generating with multimodal llms." Forty-first International Conference on Machine Learning. 2024.
> >
> > [3] Jiang, Dongzhi, et al. "CoMat: Aligning Text-to-Image Diffusion Model with Image-to-Text Concept Matching." arXiv preprint arXiv:2404.03653 (2024).
> >
> > [4] Mert Yuksekgonul, et al. “When and why vision-language models behave like bags-of-words, and what to do about it?” In The Eleventh International Conference on Learning Representations, 2022.
> >
> > [5] Patel, Maitreya, et al. "TripletCLIP: Improving Compositional Reasoning of CLIP via Synthetic Vision-Language Negatives." arXiv preprint arXiv:2411.02545 (2024).
> >
> > [6]. Le Zhang, et al. Contrasting intra-modal and ranking crossmodal hard negatives to enhance visio-linguistic fine-grained understanding. arXiv preprint arXiv:2306.08832, 2023.
> >
> > [7] Wang, et al. “Pre-trained Gaussian Processes for Bayesian Optimization.” Journal of Machine Learning Research, 2024.
> >
> > **Please let us know if you have any remaining questions. Thank you again for the insightful comments! We will ensure those points are clarified in the final version of the paper.**

---

### Official Review · Reviewer_zPFG · 2024-11-04

**Soundness:** 2
**Presentation:** 2
**Contribution:** 2
**Rating:** 5
**Confidence:** 2

**Summary:**

A new concept, build in a graph-based symbolic belief state for agents to understand its own uncertainty aboutpossible entities that might appear in the image.
It's a nice thing that  this article want to do, but  the solution is so boring. T2I + MLLM directly.

**Strengths:**

The question does arise as to how to make the generation of T2I models more personalised and more responsive to the specific and potential needs of users.
1. Design and prototypes for T2I agents that adaptively ask clarification questions and present belief states.
2. An automatic evaluation pipeline with simulated users to assess the question-asking skills of T2I agents.
3. DesignBench:a new T2I agent benchmark.

**Weaknesses:**

The solution is too simple and very uninventive. Put in a new concept and then come back with a self-explanatory statement.
Beliefstate It's a former concept, and there's nothing inherently innovative about it.

**Questions:**

Focus on D: implementation details and show your novelty. The main text is too shallow.

---

> ### Author Response · Authors · 2024-11-20
>
> We thank the reviewer for acknowledging the novelty and simplicity of our work.
>
> > It's a nice thing that this article want to do, but the solution is so boring.
>
> We urge the reviewer to follow Reviewer Guidelines (https://iclr.cc/Conferences/2019/Reviewer_Guidelines) when rating the paper. “Boring” is a subjective feeling, not a rubric to make recommendations on a research paper.
>
>  > The solution is too simple and very uninventive… Beliefstate It's a former concept, and there's nothing inherently innovative about it.
>
> We believe that a simple solution is a merit, not a disadvantage especially when dealing with HCI and User Interfaces, simplicity is essential to drive utility. For the novelty of this work, please see the general reply, where we also summarized the core differences between the conventional belief states and our graph-based T2I agent belief.
>
> >  Put in a new concept and then come back with a self-explanatory statement.
>
> Could you please clarify what you mean by this?
>
> > Focus on D: implementation details and show your novelty. The main text is too shallow.
>
> Thank you for the suggestion. We will add the pseudocode for our sequential ICL approach for belief graph generation, which summarizes Appendix D and emphasizes the novelty of the belief graph generation method. We will also update the paper to clarify other aspects of the novelty of this work detailed in the general reply.

---

### Official Review · Reviewer_m762 · 2024-11-04

**Soundness:** 3
**Presentation:** 3
**Contribution:** 3
**Rating:** 6
**Confidence:** 3

**Summary:**

This work addresses the problem of sub-optimal image generation from text-to-image generators due to under-specified or open-ended user prompts. The work proposes building proactive T2I agents that could ask clarifications questions. Furthermore, the understanding is present as a belief state visible and editable by the user. The work also proposes a scalable automated evaluation benchmark. Experimental results suggest that 90% of the human subjects found agents and the belief states useful for the T2I workflow, and the proposed agents significantly improve the VQAScores of the generation.

**Strengths:**

* In contrast to previous multi-turn T2I systems working on multi-turn user instructions, the proposed system asks question to the users for clarifications, which is a new form of interaction with the users orthogonal to previous works.
* This work proposes an evaluation pipeline that simulates users, which makes the evaluation of proactive agents human-free and much easier. This pipeline could also benefit the development of future agents that ask questions.
* The proposed DesignBench supplements COCO-Captions in the artistic images, which is a dataset that benchmarks the capabilities of text-to-image systems tailored to the needs of designers and artists.
* Writing: the paper has its messages clearly conveyed.

**Weaknesses:**

* The work frames the problem as the agent updating beliefs according to a fixed world state in the user's mind. However, the user may not have a clear idea in mind (i.e., the user may not have a pre-defined world state). In contast, the user might want to get some inspirations from the system without constraints. This use case has not been considered in the system design. This limitation might affect users such as artists. This was discussed in L534-539, but no suggestions were proposed.
* The work uses LLMs in Ag2 and Ag3 without fine-tuning. However, this work does not explore trained LLMs (VLMs) with either image data or trajectories that include asking questions. This indicates that the LLM is purely exploring in text space. The exploration might be sub-optimal since exploration on text space might be different than exploration with images in mind.
* The output of the system still might not follow user's instructions. For example, one of the generated images in Fig. 1 does not have the rabbit chasing the dog.

**Questions:**

The reviewer has a question regarding the failure case in Fig. 1:
* Is the system bottle-necked by the alignment and prompt-following capabilities of the text-to-image model or by the capabilities of the agent?

---

> ### Author Response · Authors · 2024-11-20
>
> > The work frames the problem as the agent updating beliefs according to a fixed world state in the user's mind. However, the user may not have a clear idea in mind (i.e., the user may not have a pre-defined world state). In contast, the user might want to get some inspirations from the system without constraints. This use case has not been considered in the system design. This limitation might affect users such as artists. This was discussed in L534-539, but no suggestions were proposed.
>
> This is a great question. In practice, our system does work for the case where users do not have a clear image in mind. The clarification questions and belief states offer ideas on what could be possible in the image. For example, Ag2 asks “What type of berries would you like on the cake (e.g., strawberries, blueberries, raspberries, mixed)?” in Figure 4 and shows the possible entities that could appear in Figure 14 (e.g., the “people” entity not mentioned in the prompt). These aspects can give users inspiration on what to generate. Moreover, the user can directly edit the belief graph to change the existence (or attribute values) of an entity/attribute/relation mentioned in the prompt, so that the images do not have to satisfy any constraints imposed by the prompt itself.
>
> A belief state of an autonomous agent is a distribution over “static” world states at every turn, but the world can change whenever the human user intervenes and gives the agent new observations (e.g., answers to agent questions, edits to the belief or the prompt itself). Imagine modeling the belief of a robot, the world the robot is in can change (e.g., someone pushes the robot) and the change is reflected in the observations.
>
> The limitations on static human intention stem from theoretical considerations since we are interested in mathematically defining the objective of an intelligent agent (Section A), but the actual T2I agents we built are not limited by this assumption since these agents do not directly optimize that objective. We will make this clearer in the paper.
>
> As for L534-539, this is discussing the limitation on distinguishing different types of uncertainty. We did give concrete suggestions: the agent can directly ask the user about their preference if the uncertainty is epistemic, and acknowledge the randomness if the uncertainty is aleatory. To predict these two types of uncertainty, one option is to use the Gaussian process probing methods developed in Wang et al., 2024b. The exact method and implementation is beyond the scope of a conference paper so we leave it as future work.
>
> > The work uses LLMs in Ag2 and Ag3 without fine-tuning. However, this work does not explore trained LLMs (VLMs) with either image data or trajectories that include asking questions. This indicates that the LLM is purely exploring in text space. The exploration might be sub-optimal since exploration on text space might be different than exploration with images in mind.
>
> Thank you for the suggestion. We agree that using LLMs to explore in the text space can be sub-optimal. However, it is important to note that even this provides sufficient gains over baselines, demonstrating the efficacy of our system.
>
> We agree that fine-tuning LLM/VLM on text/image trajectories that include asking questions can potentially improve the performance of the agent. A rigorous investigation is beyond the scope of this conference paper. Future work potentially includes
> 1. collecting data such as gold-standard trajectories or annotations on the quality of trajectories of human-agent conversations;
> 2.  developing approaches to fine-tune the model on multi-turn trajectories of images and text.
>
>
> We will add a discussion about this important point the reviewer raised.
>
> Please understand that a conference paper has limited space. While we would like to make our work as perfect as possible, we had to prioritize the most important aspects (see the general reply) within the page constraints.

---

> > ### Author Response · Authors · 2024-11-20
> >
> > > Question: Is the system bottle-necked by the alignment and prompt-following capabilities of the text-to-image model or by the capabilities of the agent? (Also mentioned in weakness: The output of the system still might not follow user's instructions...)
> >
> > Yes, the system is bottle-necked by the alignment and prompt-following capabilities of the text-to-image model.
> >
> > The agent prototypes we described in the paper are highly modular: the agents use frozen T2I models to generate images based on the prompts that the agent updated. Therefore the quality of the final images are directly influenced by the ability of the T2I model employed. This also means that when a better off-the-shelf T2I model becomes available, it can be directly plugged into the agents and the system will achieve better performance without any additional adaptation.
> > We would like to emphasize that the study of T2I prompt to image alignment and the study of **iterative clarification/refinement of an underspecified prompt** are two very important and orthogonal objectives. Our paper focuses on the later objective.
> >
> > Even with a highly detailed and unambiguous prompt, the system may not produce a perfect image if the underlying T2I model is inadequate.  Conversely, even with a perfect T2I model, uncertainties or ambiguities in the prompt can hinder the system's performance.  Many works aim to address the issue of T2I prompt to image alignment [1,2,3,4,5,6] but almost none for the second in the literature of T2I. Our work focuses on the prompt underspecification problem and aims to design agents that proactively ask users clarification questions and present beliefs to gather information about what the user wants to generate. The belief graph and the final prompt of our proposed agent could have accurate entities, attributes and relationships and in some cases T2I agents fail to follow parts of the prompts or create distorted assets in the images [4].
> >
> > Table 1 and Table 2 we show evaluation metrics over T2T, T2I and I2I similarity. We can see the T2T scores as a form of system ablation, since we exclude the T2I model and only perform similarity on the caption input to the T2I model. This removes any error or prior information that the T2I model adds. In Table 1 and Table 2 per the T2T metrics, we have achieved a 92%+ similarity score, and therefore a better T2I model for the agent can immediately boost the performance of the overall agent system.
> >
> > We will make this clearer in the paper.
> >
> > [1] Wu, Tsung-Han, et al. "Self-correcting llm-controlled diffusion models." Proceedings of the IEEE/CVF Conference on Computer Vision and Pattern Recognition. 2024.
> >
> > [2] Yang, Ling, et al. "Mastering text-to-image diffusion: Recaptioning, planning, and generating with multimodal llms." Forty-first International Conference on Machine Learning. 2024.
> >
> > [3] Jiang, Dongzhi, et al. "CoMat: Aligning Text-to-Image Diffusion Model with Image-to-Text Concept Matching." arXiv preprint arXiv:2404.03653 (2024).
> >
> > [4] Mert Yuksekgonul, et al. When and why vision-language models behave like bags-of-words, and what to do about it? In The Eleventh International Conference on Learning Representations, 2022.
> >
> > [5] Patel, Maitreya, et al. "TripletCLIP: Improving Compositional Reasoning of CLIP via Synthetic Vision-Language Negatives." arXiv preprint arXiv:2411.02545 (2024).
> >
> > [6]. Le Zhang, et al. Contrasting intra-modal and ranking crossmodal hard negatives to enhance visio-linguistic fine-grained understanding. arXiv preprint arXiv:2306.08832, 2023.

---

> > > ### Comment · Reviewer_m762 · 2024-12-01
> > >
> > > The reviewer would like to thank the authors for additional information. The reviewer would maintain the original positive rating.

---

### Author Response · Authors · 2024-11-20
**General reply**

We thank the reviewers for time and feedback. We are encouraged that the reviewers found our work novel (m762 “new form of interaction”, zPFG “new concept” “new T2I agent benchmark”), effective (Tosu, JLtN), reliable (Tosu), significant (JLtN), well-written (Tosu) and clear (Tosu, m762), with comprehensive evaluation (JLtN) and strong results (Tosu). Reviewers m762, JLtN and zPFG also acknowledged that DesignBench and our automated evaluation are strengths of our contribution; they are both easy to use and beneficial for future agent development.

In the following, we would like to emphasize the novelty and contributions of this work. **We use “belief graph” as a shorthand for our graph-based T2I agent belief state.** Please note that the goal of this work is to enable better human-agent collaboration for T2I generation through clarification questions and controlling the belief graph. Improving T2I models themselves, such as text-image alignment, is not the goal of our work. Novelty and contributions:

1. System design of proactive T2I agents:
    * Novel human-agent interaction modalities: Prior to our work, human users typically interact with current T2I systems by giving additional instructions or refining the prompt. To the best of our knowledge, our work is the first to propose a proactive T2I agent system that is able to ask clarification questions and present its belief graph for the user to edit.
    * Novel human-agent interaction interface:  We designed a new interface to best enable the clarification and belief graph interaction modalities. We have not seen these features in any T2I, or other generative media apps that are publicly live to date, signifying to us total uniqueness. Our human studies showed that at least 85% of raters found each component of the interface useful for their workflow, for us proving that these are both novel and useful.
    * Novel design of different T2I agents that enable the proposed interaction modalities. Please see Section 4 of the paper for the full details of the design principles and construction of those T2I agent prototypes (Ag1, Ag2, Ag3).
2. Our **belief graph significantly differs from the classic belief state** in the following ways:
    * Hardcoded predicates v.s. Automatically-generated predicates: Traditionally, constructing classic symbolic belief states requires a pre-defined set of predicates such as “on(a, b)”, “is\_red(a)”, “at\_position(robot, x, y, z)” and it is non-trivial to learn new predicates that can be used and generalized to new tasks [1, 2]. Typically the pre-defined set of predicates are written by system developers and hardcoded into classic AI systems [3, 4].
 Our belief graph does not require any pre-defined set of predicates. Instead, we propose to construct symbolic beliefs using a sequential in-context learning (ICL) method with LLMs. This method first generates a list of entities together with their descriptions conditioned on the user description of image; then, we add each entity to the context, and let the LLM generate a list of attributes and values (this step is done in parallel across entities); and finally, we add all entities to the context and let the LLM generate relations and their attributes.  Our method can be generalized across a wide range of T2I tasks and achieve high performance (see our comprehensive results on Coco, Imageinwords, DesignBench). We will include a pseudo code for this method in the paper.
    * Application to planning v.s. T2I: To the best of our knowledge, classic symbolic belief states are mostly used for robot planning, and we are the first to use symbolic beliefs to assist T2I tasks.
    * Data structure for symbolic states / belief states – Set v.s. Graph: Because of the application to planning, a symbolic world state is usually implemented and stored as a set or list of literals (atoms or negation of atoms where atoms are instantiated predicates) [5, 6, 7] so that whenever an action is applied, the agent can apply transition by adding and deleting items in the set according to the precondition and effect of the action.
For T2I tasks, it is more convenient to use a graph to represent the world state associated with an image, since entities and relations naturally form a set of nodes (entities) and edges (relations between entities). Each component of the graph can also have probabilities, making it easy to turn a world state into a belief state using the same data structure. Hence we represent T2I agent beliefs using graphs. The agent can directly update the graph for each transition instead of going through a set or list.

---

> ### Author Response · Authors · 2024-11-20
>
> 2. (continued)
>     - Interpretability and controllability: The graph structure makes our agent belief more interpretable than traditional belief states, since we can visualize and progressively disclose the graph to the human user. Moreover, each node or relation in the belief graph has associated descriptions, making it easy for the user to understand and potentially edit every component of the belief graph. In our human studies, about 85% of raters found the belief graph useful. To the best of our knowledge, **our work is the first to use the graph-based belief state for human-AI interaction**.
> 3. Automated evaluation of T2I agents: We propose a novel automated evaluation approach for T2I agents using self-play. The agent interacts with a simulated user that has access to the original image and its long caption. See Section 5.1 (and C.4) for the full details of how the simulated user is constructed. This evaluation pipeline is easy to use and can help the future development of T2I agents.
> 4. DesignBench: We envision that a significant fraction of T2I users are artists and designers, and it is important to ensure that T2I agents are evaluated for these use cases. Hence we create DesignBench, featuring photo-realism, animation and multiple styles with short and long captions. DesignBench can be directly plugged into our automated evaluation to streamline the evaluation process.
>
> We will update our manuscript to emphasize the above novelty, including changing the ambiguous name “belief state” to “belief graph” when we refer to T2I agent beliefs, and adding the pseudocode for our sequential ICL approach for belief graph generation.
>
> Again, we are making major contributions to the human-AI interaction subfield of AI/ML. Improving computer vision models is NOT the focus of our work. We urge the area chair and reviewers to take this into consideration when making the accept/reject recommendation.
>
>
> [1] Pasula HM, Zettlemoyer LS, Kaelbling LP. Learning symbolic models of stochastic domains. Journal of Artificial Intelligence Research. 2007 Jul 21;29:309-52.
>
> [2] Xia V, Wang Z, Allen K, Silver T, Kaelbling LP. Learning sparse relational transition models. International Conference on Learning Representations (ICLR), 2019.
>
> [3] https://en.wikipedia.org/wiki/Stanford_Research_Institute_Problem_Solver
>
> [4] https://en.wikipedia.org/wiki/Planning_Domain_Definition_Language
>
> [5] Y. Alkhazraji, M. Frorath, M. Grützner, M. Helmert, T. Liebetraut, R. Mattmüller, M. Ortlieb, J. Seipp, T. Springenberg, P. Stahl, and J. Wülfing. Pyperplan. https://doi.org/10.5281/zenodo.
> 3700819, 2020. URL https://doi.org/10.5281/zenodo.3700819.
>
> [6] Caelan R. Garrett, Tomás Lozano-Pérez, Leslie P. Kaelbling. PDDLStream: Integrating Symbolic Planners and Blackbox Samplers via Optimistic Adaptive Planning, International Conference on Automated Planning and Scheduling (ICAPS), 2020.
>
> [7] Caelan R. Garrett, et al. "Online replanning in belief space for partially observable task and motion problems." 2020 IEEE International Conference on Robotics and Automation (ICRA). IEEE, 2020.

---

### Author Response · Authors · 2024-11-27

Dear reviewers and AC,

We would like to highlight some important updates we have made to the paper. Please see an anonymous video linked in the abstract for a demo of our agent.

Emphasis on novelty:
- clarified novelty in the introduction (last paragraph).
- added pseudo code for our new belief graph generation approach in Algorithm 1.
- clarified the difference between standard belief states and our belief graph in Section 4.2 and Appendix 1.

New experiments:
- We added results for validation for the ground truth caption.
- We added empirical results for ImageInWords to further enrich the evaluation.

We also further improved clarity throughout the paper according to the suggestions made by the reviewers.

-Authors

---

> ### Author Response · Authors · 2024-12-01
>
> Dear reviewers and AC,
>
> The discussion period is ending soon. Please let us know if you have any remaining comments. We really appreciate your time and effort.
>
> -Authors

---

### Author Response · Authors · 2024-12-03
**Anonymous link to video demo for further clarification**

Dear reviewers and AC,

To ensure a thorough understanding, we have put together an anonymous short video to demonstrate the UI and behaviors of our proactive T2I agents: **https://youtu.be/OitADTsaqM0**.

We hope this video can further clarify the significant contribution of our work, and make this paper more accessible to all readers.

-Authors

---

### Meta-Review · Area_Chair_jCYX · 2024-12-18

**Metareview:**

a) The paper introduces proactive T2I agents that clarify user intent via multi-turn interactions, asking questions and visualizing beliefs through an interpretable graph. Human studies and automated benchmarks show up to 2x higher VQAScore than single-turn T2I systems.

(b) Strengths:
1. Novel human-agent interaction with belief graph for transparency.
2. Comprehensive evaluation (DesignBench, COCO, ImageInWords).

(c) Weaknesses:
1. Limited exploration of fine-tuned vision-language models.
2. Bottlenecks from frozen T2I models.
(d) Decision: Reject. While the work explores a relevant issue, it fails to address modern image-editing advancements, and relies on incremental improvements.

**Additional Comments On Reviewer Discussion:**

Reviewers raised concerns about limited novelty, reliance on frozen T2I models, and lack of comparison with advanced image-editing methods. The authors clarified their focus on prompt underspecification and emphasized the belief graph’s interpretability. While responses were detailed, fundamental weaknesses—incremental contribution and omission of key comparisons—remain.

---

### Decision · Program_Chairs · 2025-01-22

Reject